# The most active Cu facet for low-temperature water gas shift reaction

Zhenhua Zhang[1], Sha-Sha Wang[2], Rui Song[1], Tian Cao[1], Liangfeng Luo[1], Xuanye Chen[3], Yuxian Gao[1], Jiqing Lu[3], Wei-Xue Li[1,2] & Weixin Huang [1]

Identification of the active site is important in developing rational design strategies for solid catalysts but is seriously blocked by their structural complexity. Here, we use uniform Cu nanocrystals synthesized by a morphology-preserved reduction of corresponding uniform $Cu_2O$ nanocrystals in order to identify the most active Cu facet for low-temperature water gas shift (WGS) reaction. Cu cubes enclosed with {100} facets are very active in catalyzing the WGS reaction up to 548 K while Cu octahedra enclosed with {111} facets are inactive. The Cu–Cu suboxide ($Cu_xO$, $x \geq 10$) interface of Cu(100) surface is the active site on which all elementary surface reactions within the catalytic cycle proceed smoothly. However, the formate intermediate was found stable at the Cu–$Cu_xO$ interface of Cu(111) surface with consequent accumulation and poisoning of the surface at low temperatures. Thereafter, Cu cubes-supported ZnO catalysts are successfully developed with extremely high activity in low-temperature WGS reaction.

[1] Hefei National Laboratory for Physical Sciences at Microscale, CAS Key Laboratory of Materials for Energy Conversion and Department of Chemical Physics, University of Science and Technology of China, Hefei 230026, China. [2] Dalian Institute of Chemical Physics, Chinese Academy of Sciences, University of Chinese Academy of Sciences, Dalian 116023, China. [3] Key Laboratory of the Ministry of Education for Advanced Catalysis Materials, Institute of Physical Chemistry, Zhejiang Normal University, Jinhua 321004, China. Zhenhua Zhang and Sha-Sha Wang contributed equally to this work. Correspondence and requests for materials should be addressed to W.H. (email: huangwx@ustc.edu.cn)

Catalyst nanoparticles (NPs) generally expose a variety of surface sites (e.g., facets, steps, and corners), each with their distinct reactivity. Such structural complexity seriously impairs efforts to identify their active site for developing rational design strategies in heterogeneous catalysis. The complexity of catalyst NPs has been traditionally simplified by the use of single crystals with well-defined surface structures as model catalysts[1]. Recently uniform catalyst nanocrystals (NCs) with tunable structures have demonstrated great potential in heterogeneous catalysis either as a novel type of model catalysts that can be studied under the same conditions as powder catalysts or as excellent candidates as efficient catalysts[2–5]. However, capping ligands on catalyst NCs inherited from the wet chemistry synthesis strongly affect their applications in heterogeneous catalysis, particularly in gas-solid heterogeneous catalytic reactions[6–8].

The water gas shift (WGS) reaction is commonly used in the chemical industry for the production of clean $H_2$[9]. It is mildly exothermic and thermodynamically favors low reaction temperatures. Cu-based catalysts are currently used industrially for low-temperature WGS reaction. Thus it is of great importance to fundamentally understand the structure-activity relation of Cu-based catalysts for optimizing the catalyst structure. The WGS reaction were studied over Cu (100), (110), and (111) single-crystal surfaces[10, 11], and the results demonstrated its structure-sensitivity.

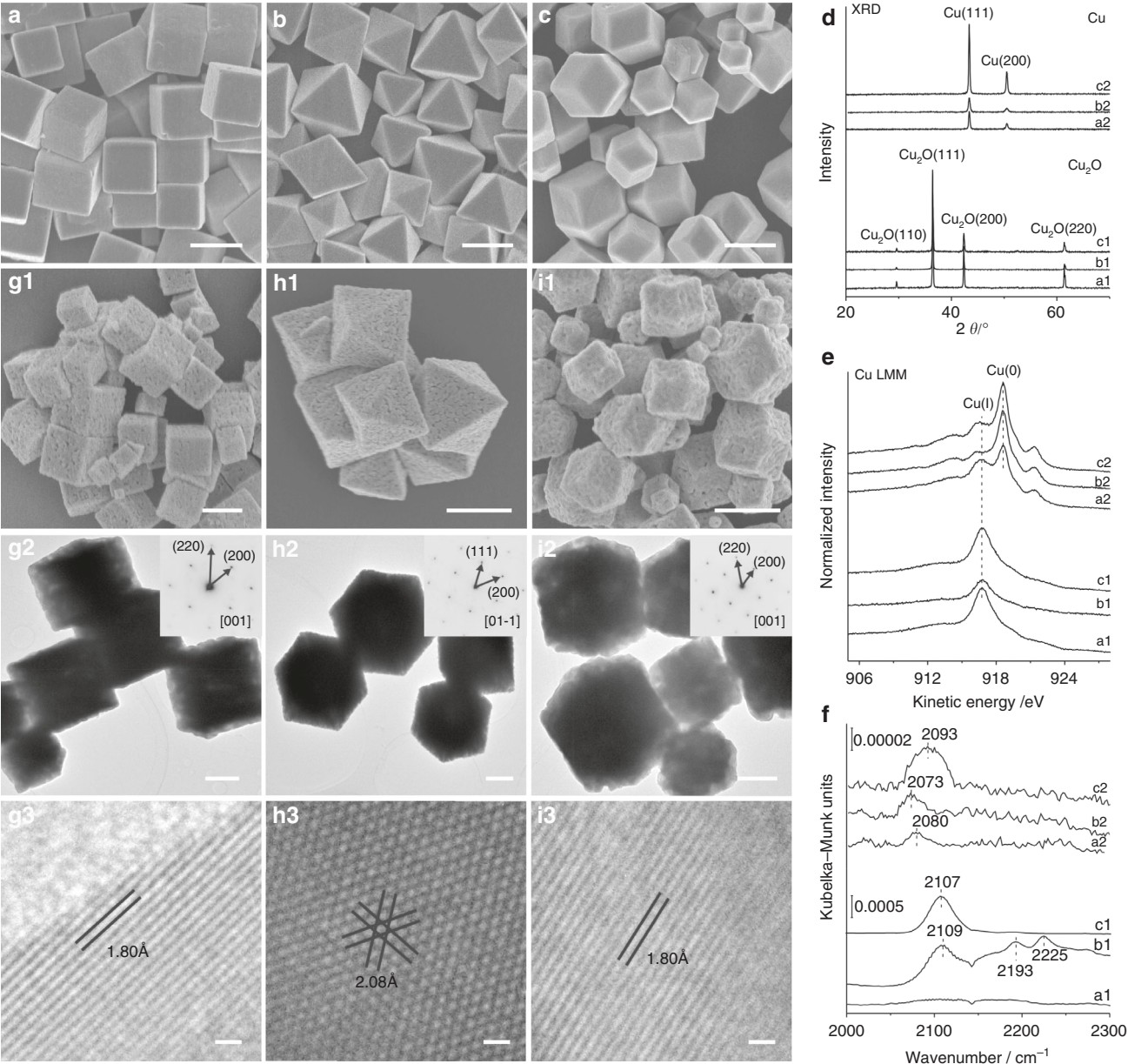

**Fig. 1** Structural characterizations. The *scale bars* of **a**–**c** and (**g1**–**i1**) correspond to 1000 nm, that of (**g2**–**i2**) correspond to 500 nm, and that of (**g3**–**h3**) correspond to 2 nm. Representative SEM images of **a** $Cu_2O$ cubes, **b** $Cu_2O$ octahedral, and **c** $Cu_2O$ rhombic dodecahedra. **d** XRD patterns, **e** Cu LMM AES spectra measured without exposure to air, and **f** in situ DRIFTS spectra of CO adsorption at 123 K of $Cu_2O$ cubes (a1), octahedra (b1), rhombic dodecahedra (c1) and Cu cubes (a2), octahedra (b2) and rhombic dodecahedra (c2). Representative SEM, TEM and HRTEM images of (**g1**–**g3**) Cu cubes, (**h1**–**h3**) Cu octahedra and (**i1**–**i3**) Cu rhombic dodecahedra. The insets in the HRTEM images show the ED patterns of corresponding Cu nanocrystals. The lattice fringes of 1.80 and 2.08 Å, respectively, correspond to the spacing of Cu{200} and {111} crystal planes (JCPDS card NO. 89-2838)

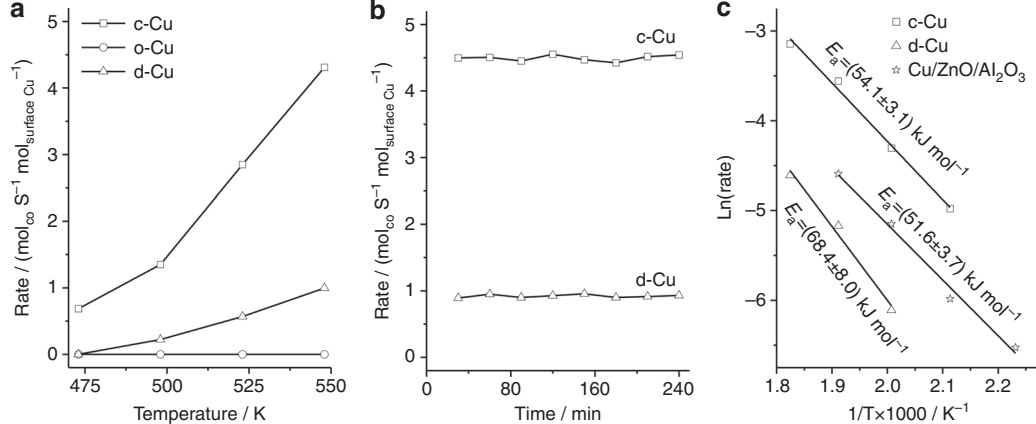

**Fig. 2** Catalytic performance. **a** Reaction rate ($mol_{CO}$ $s^{-1}$ $mol_{surface\ Cu}^{-1}$) of Cu cubes (c-Cu), octahedra (o-Cu), and rhombic dodecahedra (d-Cu) in the WGS reaction as a function of reaction temperature. **b** Reaction rate of Cu cubes (c-Cu) and rhombic dodecahedra (d-Cu) at 548 K in the WGS reaction as a function of reaction time. **c** Arrhenius plots of Cu cubes (c-Cu) and rhombic dodecahedra (d-Cu) and commercial $Cu/ZnO/Al_2O_3$ catalyst in the WGS reaction

Here, we report the identification of the most active Cu facet for low-temperature WGS reaction up to 548 K employing uniform capping ligands-free Cu NCs synthesized by a morphology-preserved reduction of corresponding uniform $Cu_2O$ NCs. Cubic Cu NCs enclosed with {100} facets are more catalytically active than dodecahedral Cu NCs enclosed with {110} facets while octahedral Cu NCs enclosed with {111} facets are inactive. The Cu–Cu suboxide ($Cu_xO$, $x \geq 10$) interface of Cu (100) surface is the active site on which all elementary surface reactions within the catalytic cycle proceed smoothly, but the Cu–$Cu_xO$ interface of Cu(111) surface is poisoned by an accumulation of formate intermediate stable at low reaction temperatures. Thereafter, we successfully developed Cu cubes-supported ZnO catalysts with extremely high activity in low-temperature WGS reaction.

## Results

**Synthesis and structures of Cu NCs**. A morphology-preserved reduction strategy was firstly developed to synthesize uniform Cu NCs from corresponding $Cu_2O$ NCs. Uniform capping ligands-free $Cu_2O$ cubes (denoted as c-$Cu_2O$), octahedra (denoted as o-$Cu_2O$) and rhombic dodecahedra (denoted as d-$Cu_2O$) respectively enclosed with {100}, {111} and {110} facets were synthesized following our previously established procedures[12]. Their structures were confirmed by both microscopic and spectroscopic characterizations (Fig. 1a–f, Supplementary Figs. 1 and 2). The size distribution of c-$Cu_2O$, o-$Cu_2O$, and d-$Cu_2O$ is $1000 \pm 150$, $1056 \pm 207$, and $595 \pm 113$ nm, respectively (Supplementary Fig. 3). According to the CO-temperature programmed reduction (CO-TPR) spectra (Supplementary Fig. 4), a reduction process in 5% CO/Ar at 548 K for 2 h was chosen to reduce $Cu_2O$ NCs. After CO reduction, the XRD patterns change from those of starting $Cu_2O$ (JCPDS card NO. 78-2076) completely into those of metallic Cu (JCPDS card NO. 89-2838) (Fig. 1d) while the scanning electron microscopy (SEM), transmission electron microscopy (TEM), and high-resolution transmission electron microscopy (HRTEM) images (Fig. 1g–i) show a well preservation of the morphologies of starting $Cu_2O$ NCs. The electron diffraction (ED) patterns confirm that all acquired Cu NCs are single crystals. Thus, uniform $Cu_2O$ cubes, octahedra, and rhombic dodecahedra can be reduced respectively into uniform Cu cubes (denoted as c-Cu), octahedra (denoted as o-Cu), and rhombic dodecahedra (denoted as d-Cu). Such a morphology-preserved reduction of $Cu_2O$ NCs into Cu NCs can

be attributed to the cubic phase structures of both $Cu_2O$ and Cu, the not too large difference between the lattice constants of $Cu_2O$ ($a = 426.7$ pm) and Cu ($a = 361.5$ pm), the large sizes of $Cu_2O$ NCs, and the low reduction temperature. However, the acquired Cu NCs exhibit rough surfaces likely resulting from the lattice mismatch between $Cu_2O$ and Cu. The acquired Cu NCs are finer than the corresponding $Cu_2O$ NCs, and the size distribution of c-Cu, o-Cu, and d-Cu is $877 \pm 157$, $937 \pm 216$, and $497 \pm 137$ nm, respectively (Supplementary Fig. 3). Consequently, c-Cu, o-Cu, and d-Cu exhibit larger specific BET surface areas, respectively, of 1.17, 2.35, and 3.33 $m^2$ $g^{-1}$ than the corresponding c-$Cu_2O$, o-$Cu_2O$, and d-$Cu_2O$, respectively, of 0.67, 1.58, and 1.98 $m^2$ $g^{-1}$. Although with a larger average edge length, the o-$Cu_2O$ and o-Cu NCs exhibit larger specific BET surface areas, respectively, than the c-$Cu_2O$ and c-Cu NCs due to the morphology and facet effects. The surface area-to-volume ratio of an octahedron and a cube can be geometrically calculated, respectively, as $3\sqrt{6}/L$ and $6/L$ ($L$ being the edge length); meanwhile, the surface atom density of the {111} facets exposed on octahedral NCs is higher than that of the {100} facets exposed on cubic NCs for $Cu_2O$ and Cu with cubic phases.

The acquired Cu NCs were characterized by x-ray photoelectron spectroscopy (XPS) without exposure to air. The Cu LMM Augur electron spectra (AES) (Fig. 1e) and Cu 2p XPS spectra (Supplementary Fig. 5) demonstrate dominance of metallic Cu on surfaces of all acquired Cu NCs; however, the Cu(I) feature is still visible. In the corresponding O 1s and C 1s XPS spectra (Supplementary Fig. 5), all Cu NCs exhibit three O 1s components at 529.5, 530.3, and 531.8 eV and three C 1s components at 284.8, 289.0, and 287.7 eV while o-Cu NCs exhibit an additional minor C 1s component at 286.6 eV. The O 1s components at 529.5 and 531.8 eV could be respectively assigned to oxygen adatoms and hydroxyl groups/oxygenates species while the O 1s component at 530.3 eV was observed to vary simultaneously with the Cu(I) feature for all acquired Cu NCs and, thus, could be assigned to Cu suboxide ($Cu_xO$, $x \geq 10$)[13, 14]. The C 1s components at 284.8, 289.0, 287.7, and 286.6 eV could be assigned respectively to adventitious carbon, carbonate, carboxylate and formate species[15]. In situ diffuse reflectance infrared Fourier transformed spectra (DRIFTS) of the reduction processes of $Cu_2O$ NCs into Cu NCs (Supplementary Fig. 6) show significant attenuations of $Cu_2O$ vibrational features (652, 794, and 1123 $cm^{-1}$)[16] and formations of carboxylate (1277 $cm^{-1}$) and carbonate (1235, 1455 and 1506 $cm^{-1}$) species on all Cu NCs and

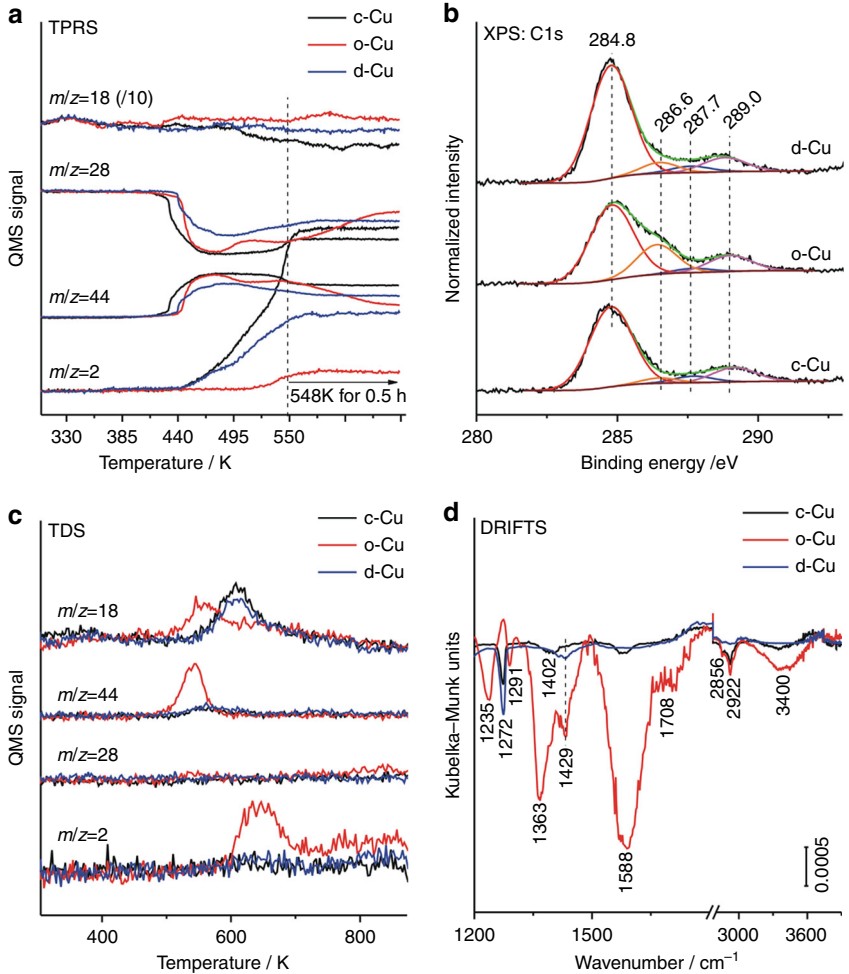

**Fig. 3** Reaction mechanism. **a** Temperature-programmed reaction spectra of WGS reaction over Cu cubes (c-Cu), octahedra (o-Cu), and rhombic dodecahedra (d-Cu). **b** C 1 s XPS spectra of Cu cubes (c-Cu), octahedra (o-Cu), and rhombic dodecahedra (d-Cu) after WGS reaction at 548 K measured without exposure to air. **c** Thermal desorption spectra of Cu cubes (c-Cu), octahedra (o-Cu), and rhombic dodecahedra (d-Cu) after WGS reaction at 548 K. **d** In situ diffuse reflectance infrared spectra of Cu cubes (c-Cu), octahedra (o-Cu), and rhombic dodecahedra (d-Cu) after WGS reaction at 548 K followed by heating in Ar to 723 K

formate species (1588 cm$^{-1}$)[17, 18] exclusively on o-Cu NCs, in consistence with the above XPS results. All these observed surface species on acquired Cu NCs are common on metal powder catalysts.

Figure 1f compares in situ DRIFTS spectra of CO adsorption on Cu$_2$O and Cu NCs at 123 K. Agreeing with previous reports[12, 19–21], a vibrational band at ~2108 cm$^{-1}$ was observed on o-Cu$_2$O and d-Cu$_2$O and assigned to CO-adsorbed at the Cu (I) sites exposed on {111} and {110} facets while no feature was observed on c-Cu$_2$O enclosed with O-terminated Cu$_2$O (100) facets. Additional vibrational bands at ~2193 and 2225 cm$^{-1}$ were observed on o-Cu$_2$O and assigned to CO$_2$ adsorbed at the coordination-unsaturated Cu(I) sites of {111} facets formed by the reaction of CO with adsorbed oxygen species[12, 19]. Vibrational bands of CO adsorbed on Cu NCs are diffuse and weak, and depend on their morphologies. It is located at 2080 cm$^{-1}$ on c-Cu, 2073 cm$^{-1}$ on o-Cu, and 2093 cm$^{-1}$ on d-Cu that respectively correspond well to that of CO adsorbed on Cu (100), (111), and (110) single-crystal surfaces[22–25]. No feature of CO adsorbed at the Cu(I) site could be observed for all Cu NCs, indicating that Cu$_x$O on Cu NCs is O-terminated. Absence of vibrational features above 2100 cm$^{-1}$ corresponding to CO adsorbed at stepped sites and defects on Cu surfaces[26] suggests low densities of stepped sites and defects on the surfaces of all acquired Cu NCs. These

CO adsorption results demonstrate that although with rough surfaces, uniform c-Cu, o-Cu, and d-Cu NCs dominantly expose {100}, {111}, and {110} terrace Cu sites on the surfaces, respectively. Therefore, the above comprehensive characterization results demonstrate a successful synthesis of uniform c-Cu, o-Cu, and d-Cu NCs respectively with well-defined {100}, {111}, and {110} surfaces via a morphology-preserved reduction of corresponding c-Cu$_2$O, o-Cu$_2$O, and d-Cu$_2$O NCs.

**Effect of Cu facet on activity**. Various types of Cu NCs exhibit morphology-dependent catalytic activity in the WGS reaction. CO conversion over the c-Cu and d-Cu NCs was observed, respectively, at 473 and 498 K but not for the o-Cu NCs up to 548 K (Supplementary Fig. 7). Catalytic activity of commercial Cu/ZnO/Al$_2$O$_3$ catalyst in the WGS reaction was also evaluated (Supplementary Fig. 8). On the basis of surface Cu atoms derived from the specific surface areas of c-Cu and d-Cu NCs and the Cu atom densities of Cu (100) and (110) surfaces, the calculated surface Cu atom-specific reaction rates of c-Cu NCs are much higher than those of d-Cu NCs (Fig. 2a). The c-Cu and d-Cu NCs are also catalytically stable at 548 K (Fig. 2b and Supplementary Fig. 7). SEM images (Supplementary Fig. 9) demonstrate that the Cu NCs preserve their morphologies after the activity evaluation

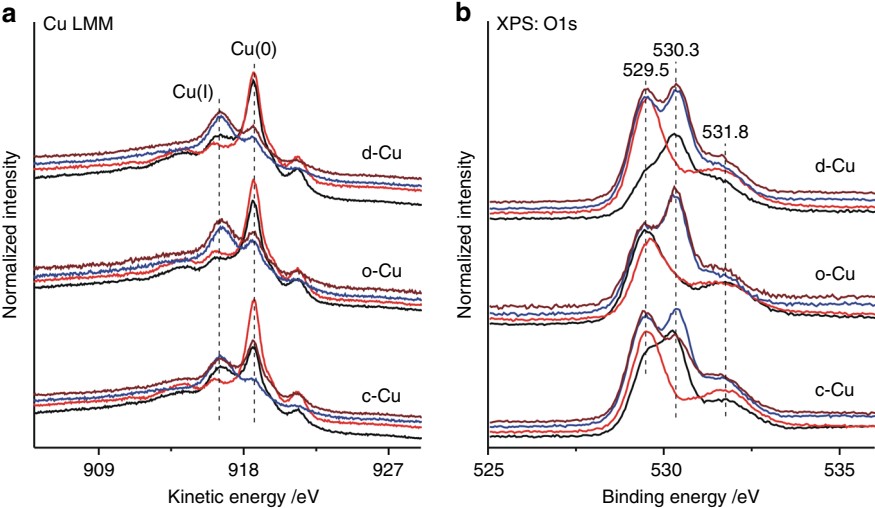

**Fig. 4** Water activation. **a** Cu LMM AES and **b** O 1s XPS spectra of various Cu NCs (*black lines*), Cu NCs exposed to $H_2O$ at 423 K (*red lines*), Cu NCs exposed to $H_2O$ at 523 K (*blue lines*) and Cu NCs exposed to $H_2O$ at 523 K and then to CO at 473 K (*brown lines*). All spectra were measured without exposure to air

up to 548 K. Results of a Weisz–Prater analysis and a Mears analysis[27] (Supplementary Note 1, 2, and 3) show the absence of mass and heat transfer limitations under our reaction conditions. Thus the Arrhenius plots (Fig. 2c) were made to calculate the apparent activation energy ($E_a$) of WGS reaction catalyzed by c-Cu, d-Cu, and Cu/ZnO/Al$_2$O$_3$, respectively, as $54.1 \pm 3.1$, $68.4 \pm 8.0$, and $51.6 \pm 3.7$ kJ mol$^{-1}$. Therefore, employing various types of Cu NCs, we demonstrate an obvious facet effect of Cu particles in low-temperature WGS reaction with the Cu{100} facet as the most active facet.

It is noteworthy that the $E_a$ value of Cu/ZnO/Al$_2$O$_3$ is similar to the previously reported value (53 kJ mol$^{-1}$)[28] while the $E_a$ values of c-Cu and d-Cu NCs differ respectively from those of Cu (100) (63.5 kJ mol$^{-1}$)[11] and Cu (110) (41.8 kJ mol$^{-1}$)[10] single-crystal surfaces. Moreover, the Cu(111) single-crystal surface was reported to exhibit an $E_a$ of 71.1 kJ mol$^{-1}$ in WGS reaction[10] while o-Cu NCs are inactive under our reaction condition. Catalytic activity of Cu single-crystal surfaces were evaluated under the very different reaction conditions[10, 11] from ours, for example, the adopted temperatures were between 573 and 673 K. We also evaluated catalytic activity of c-Cu and o-Cu in WGS reaction up to 673 K (Supplementary Fig. 10). o-Cu becomes active at 623 K, and c-Cu exhibits CO conversions that increase with the reaction temperature up to 623 K but then decrease. The results of recycling measurements of catalytic activity and SEM images of Cu NCs after the activity evaluation clearly demonstrate that both Cu NCs are not stable and undergo obvious morphological changes during the activity evaluation up to 673 K.

**In situ restructuring of Cu$_2$O NCs into Cu NCs**. We found that, when directly used as catalysts for WGS reaction, various types of Cu$_2$O NCs exhibited the steady-state catalytic activity and calculated apparent activation energy almost identical to those of corresponding Cu NCs (Supplementary Fig. 11). Structural characterization results (Supplementary Figs. 12–14) demonstrate that Cu$_2$O NCs undergo an in situ morphology-preserved reduction into corresponding Cu NCs with the presence of both Cu(0) and Cu(I) on their surfaces during WGS reaction up to 548 K. These results demonstrate that Cu NCs with co-existing Cu(0) and Cu(I) species on the surfaces are active in catalyzing

low-temperature WGS reaction and further support that the Cu {100} facet is the most active facet.

**Reaction mechanism**. Figure 3a presents temperature-programmed reaction spectra of WGS reaction over Cu NCs. The CO$_2$ production was observed to occur prior to the H$_2$ production, suggesting the H$_2$ production as the rate-limiting step in the Cu-catalyzed WGS reaction. Both CO$_2$ and H$_2$ productions proceed most facilely over c-Cu, in consistence with its highest catalytic activity. o-Cu exhibits a higher initial CO$_2$ production than d-Cu, but a much more difficult H$_2$ production. With the WGS reaction proceeding at 548 K, c-Cu and d-Cu capable of catalyzing both CO$_2$ and H$_2$ productions exhibit stable activity, respectively, while o-Cu capable of catalyzing the CO$_2$ production but few H$_2$ production gets gradually poisoned. These observations suggest that the inactivity of o-Cu in WGS reaction up to 548 K should result from self-poisoning due to the formation and accumulation of stable hydrogen-containing intermediates on the surface.

Various used Cu NCs catalysts after WGS reaction at 548 K were characterized by XPS without exposure to air. As shown in the C 1s XPS spectra (Fig. 3b), in addition to the C 1s components of adventitious carbon (284.8 eV), carboxylate (287.7 eV), and carbonate (289.0 eV), the C 1s component of formate species (286.6 eV) absent on the fresh c-Cu and d-Cu catalysts appears on the used c-Cu and d-Cu catalysts after the WGS reaction and its intensity greatly increases on the used o-Cu catalyst as compared to that on the fresh o-Cu catalyst. In the corresponding Cu LMM AES spectra, Cu 2p, and O 1s XPS spectra (Supplementary Fig. 15), no Cu(II) XPS feature appears on all used Cu catalysts and the Cu LMM AES spectra and O 1s XPS spectra of the used o-Cu catalyst change very slightly, but the metallic Cu LMM AES component increases at the expense of the Cu(I) LMM AES component on the used c-Cu and d-Cu catalysts, and accordingly, the O 1s components of oxygen adatoms (529.5 eV) and hydroxyl groups/oxygenates species (531.8 eV) increase at the expense of the O 1s component of the Cu$_x$O species (530.3 eV). Thus, during the WGS reaction up to 548 K, the copper speciation does not vary much on the inactive o-Cu catalyst surface but an accumulation of formate species obviously occurs, and the transformation of Cu$_x$O species

into the Cu(0) and oxygen adatoms, together with the formation of hydroxyl groups and oxygenates species (including the formate species), occurs on the active c-Cu and d-Cu surfaces.

The surface species on various used Cu NCs catalysts after WGS reaction at 548 K were further probed by temperature-programmed desorption spectra (TDS) in Ar and in situ DRIFTS spectra. In the TDS spectra in Ar (Fig. 3c), the used c-Cu and d-Cu catalysts exhibit a weak and broad $CO_2$ desorption peak at 560 K and a strong $H_2O$ desorption peak at 610 K while the used o-Cu catalyst exhibits strong $CO_2$ and $H_2O$ desorption peaks simultaneously at 540 K and $H_2$, and $H_2O$ desorption peaks simultaneously at 640 K. In the corresponding in situ DRIFTS spectra (Fig. 3d), the used c-Cu and d-Cu catalysts exhibit weak attenuation of vibrational peaks of carboxylate (1272–1291 cm$^{-1}$), carbonate (1235 and 1402–1429 cm$^{-1}$), formate (1363, 1588, 2856, and 2922 cm$^{-1}$), and hydroxyl (1708 and 3400 cm$^{-1}$) species[17, 18] while the used o-Cu catalyst exhibits strong loss of vibrational peaks of formate and hydroxyl species and weak attenuation of vibrational peaks of carboxylate and carbonate species. Therefore, upon heating, the reactions of carboxylate, carbonate and formate species on the used c-Cu and d-Cu catalysts produce minor $CO_2$ at 560 K and the hydroxyl groups react each other to predominantly produce water at 610 K while the reaction between co-adsorbed formate species and hydroxyl group on the used o-Cu catalyst produce much $CO_2$ and $H_2O$ at 540 K[29] and the hydroxyl groups react each other to simultaneously produce $H_2$ and $H_2O$ desorption peaks at 640 K.

The above comprehensive characterization results demonstrate accumulations of formate and hydroxyl intermediates during the WGS reaction up to 548 K on the o-Cu surface but not on the c-Cu and d-Cu surfaces that should lead to the self-poisoning and subsequent inactivity of o-Cu NCs. It is noteworthy that the surface reaction between co-adsorbed formate and hydroxyl to produce $CO_2$ and $H_2O$ at 540 K observed on the used o-Cu catalyst should be thermodynamically quenched during WGS reaction with $H_2O$ as a reactant. Meanwhile, the accumulation of formate and hydroxyl intermediates on the o-Cu surface could be alleviated at higher reaction temperatures, as the o-Cu NCs are active in catalyzing the WGS reaction at 623 K and above.

Activation of $H_2O$ on various Cu NCs, a key step in WGS reaction, was studied by XPS without exposure to air (Fig. 4, Supplementary Figs. 16, 17, and Supplementary Table 1). $H_2O$ dissociation was observed to occur on the c-Cu and d-Cu surfaces at 423 K but barely on the o-Cu surface, resulting in the growth of both hydroxyl group and oxygen adatoms at the expense of Cu$_x$O. This suggests the Cu–Cu$_x$O interface as the active site for the $H_2O$ dissociation into the hydroxyl group accompanied by the decomposition of Cu$_x$O into oxygen adatoms. In the corresponding Cu LMM AES spectra, the decomposition of Cu$_x$O on the c-Cu and d-Cu surfaces at 423 K is also evidenced by the growth of Cu(0) at the expense of Cu(I), and this process is more extensive on the c-Cu surface than on the d-Cu surface. $H_2O$ dissociates on all Cu NCs at 523 K, leading to the growth of oxygen adatoms and Cu$_x$O at the expense of hydroxyl group, indicating the decomposition of hydroxyl group into oxygen adatom that can further transform into Cu$_x$O.

The Cu NCs subjected to $H_2O$ dissociation at 523 K were then exposed to CO at 473 K. Both Cu LMM AES spectra and O 1s XPS spectra (Fig. 4, Supplementary Fig. 17 and Supplementary Table 1) demonstrate a reaction of Cu$_x$O with CO on the c-Cu surface, but not on the d-Cu and o-Cu surfaces. Thus Cu$_x$O on the Cu surface is an active oxygen species for the $CO_2$ production during WGS reaction while the co-existing oxygen adatoms serve as an oxygen reservoir for the formation

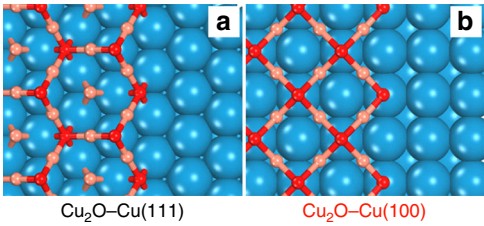

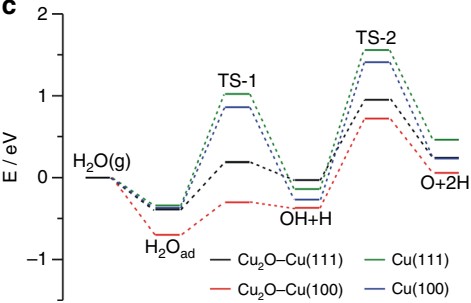

**Fig. 5** DFT calculations. Schematic **a** Cu$_2$O–Cu(111) and **b** Cu$_2$O–Cu(100) interface structures. *Blue*, *orange*, and *red balls* represent Cu, Cu of Cu$_2$O, and O atoms, respectively. **c** Calculated energy diagram of water adsorption and decomposition on Cu$_2$O–Cu(111), Cu$_2$O–Cu(100), Cu(111), and Cu(100) surfaces

and decomposition of Cu$_x$O. Since CO adsorbs on the metallic Cu surface instead of the Cu$_x$O surface (Fig. 1f), the reaction of Cu$_x$O with CO on the c-Cu surface should proceed at the Cu–Cu$_x$O interface. The reactivity of Cu$_x$O at the Cu–Cu$_x$O interface toward CO is higher on the c-Cu surface than on the d-Cu and o-Cu surfaces.

Therefore, the Cu–Cu$_x$O interface on the Cu surface capable of both dissociating water and reacting with CO should be the active site to catalyze the WGS reaction. The Cu–Cu$_x$O interface on the c-Cu surface is more active in both water dissociation and $CO_2$ production than that on the d-Cu and o-Cu surfaces, consequently, the c-Cu NCs is more catalytic active in the WGS reaction than the d-Cu and o-Cu Cu NCs.

The mechanisms of morphology-dependent catalytic performance of Cu NCs in WGS reaction were further explored by DFT calculations employing Cu(100) and Cu(111) surfaces with Cu$_2$O islands to respectively model the c-Cu and o-Cu surfaces with Cu$_x$O (Fig. 5a, b). According to previous results[30–33], ring-like structures were adopted for Cu$_2$O islands on both Cu(100) and Cu(111) surfaces. This gives a zig-zag chain structure of Cu$_2$O at the Cu–Cu$_2$O interface that was used during simulations of reactions at the Cu–Cu$_2$O interface. In consistence with previous reports[34, 35], the dissociation of water on Cu(100) and Cu(111) into OH group and H adatom needs to overcome barriers respectively of 1.23 and 1.36 eV, and the further dissociation of OH group into O and H adatoms on Cu(100) and Cu(111) needs to overcome barriers respectively of 1.68 and 1.70 eV (Fig. 5c, Supplementary Figs. 18, 19 and Supplementary Table 2); however, the dissociation of water at the Cu–Cu$_2$O interface on Cu(100) and Cu(111) into a OH group on Cu surface (OH$_{Cu}$) and a H adatom at the O of Cu$_2$O (O$_{Cu_2O}$H) proceeds facilely with barriers respectively of 0.40 and 0.58 eV, and the H of O$_{Cu_2O}$H can migrate to the neighboring Cu atom to form a H adatom on the Cu surface (O$_{Cu_2O}$H → O$_{Cu_2O}$ + H$_{Cu}$) with activation energies of 1.09 and 0.98 eV at the Cu–Cu$_2$O interface respectively on Cu(100) and Cu(111) surfaces. Thus the water activation is much more facile at the Cu$_2$O–Cu interface than at the Cu surface, and the water dissociation at the Cu–Cu$_2$O interface into OH$_{Cu}$ and O$_{Cu_2O}$H should proceed more facilely

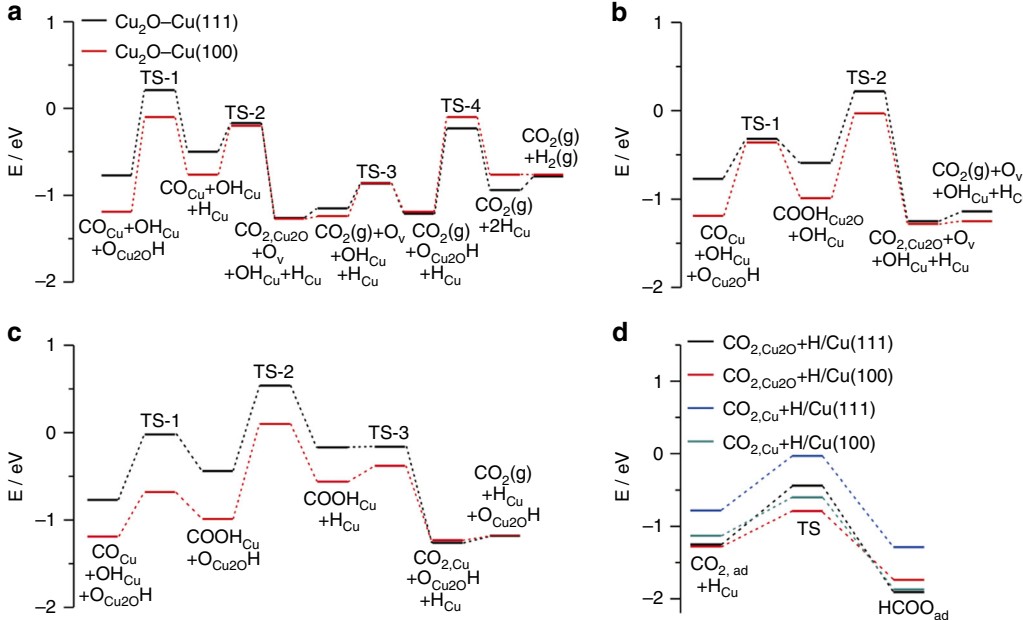

**Fig. 6** Calculated energy diagram of WGS reaction. **a** The reaction for CO at the Cu site ($CO_{Cu}$) and O of $Cu_2O$ at $Cu_2O$–Cu(111) (*black line*) and $Cu_2O$–Cu (100) (*red line*) interfaces. $O_v$ represents an oxygen vacancy in $Cu_2O$. **b** The reaction for $CO_{Cu}$ and $O_{Cu2O}H$ at the $Cu_2O$–Cu(111) (*black line*) and $Cu_2O$–Cu (100) (*red line*) interfaces. **c** The reaction for $CO_{Cu}$ and $OH_{Cu}$ at the $Cu_2O$–Cu(111) (*black line*) and $Cu_2O$–Cu(100) (*red line*) interfaces; the subsequent elementary reactions following the formation of $CO_2(g)$ + $O_v$ + $OH_{Cu}$ + $H_{Cu}$ are identical to those in **a**. **d** The elementary reaction between $CO_{2,ad}$ and $H_{Cu}$ to form $HCOO_{ad}$ at the $Cu_2O$-Cu(111) and $Cu_2O$-Cu(100) interfaces

on Cu(100) than on Cu(111). These agree with the above experimental observations (Fig. 4).

Following the water dissociation, CO adsorbed at the Cu site ($CO_{Cu}$) can react with O of $Cu_2O$ at the Cu–$Cu_2O$ interface to complete the WGS reaction cycle via the redox mechanism (Fig. 6a, Supplementary Figs. 18, 20 and Supplementary Table 3); $CO_{Cu}$ can also react either with $O_{Cu2O}H$ to produce $COOH_{Cu2O}$ (Fig. 6b, Supplementary Figs. 18, 21 and Supplementary Table 4) or with $OH_{Cu}$ to produce $COOH_{Cu}$ (Fig. 6c, Supplementary Figs. 18, 22 and Supplementary Table 5) to complete the WGS reaction cycle via the associative mechanism. All three pathways can proceed at the Cu–$Cu_2O$ interface of Cu(100) and Cu(111) surfaces with similar largest activation energies of around 1 eV. In all three pathways the elementary surface reaction with the largest activation energy is the reaction of $O_{Cu2O}H \rightarrow O_{Cu2O}$ + $H_{Cu}$ involved for the $H_2$ production. This agrees with the experimental observations that the $H_2$ production is the rate-limiting step in the Cu-catalyzed WGS reaction (Fig. 3a).

Besides producing $H_2$ and $CO_2$, the formed $H_{Cu}$ and $CO_{2,ad}$ at the Cu–$Cu_2O$ interface can facilely react to form the formate species on both Cu(111) and Cu(100) surfaces (Fig. 6d, Supplementary Figs. 18, 23 and Supplementary Tables 3–5), in consistence with previous reports[34]. The barriers of the decomposition of $HCOO_{Cu}$ species into $CO_{2,Cu}$ + $H_{Cu}$ at the Cu–$Cu_2O$ interfaces of Cu(111) and Cu(100) surfaces are similar, suggesting their similar stabilities; however, the barrier of the decomposition of $HCOO_{Cu2O}$ species into $CO_{2,Cu2O}$ + $H_{Cu}$ is respectively 1.47 and 0.95 eV at the Cu–$Cu_2O$ interfaces of Cu(111) and Cu (100) surfaces, suggesting a much higher stability of $HCOO_{Cu2O}$ species on Cu(111) than on Cu(100). Thus, it can be expected that the formate species will accumulate at the Cu–$Cu_2O$ interface of Cu(111) surface due to the large barrier for $HCOO_{Cu2O}$ decomposition during low-temperature WGS reaction but not at the Cu–$Cu_2O$ interface of Cu(100) surface, which will eventually block the active Cu–$Cu_2O$ interface of Cu(111) surface. This is consistent with the experimental observations

that the c-Cu catalyst is active in catalyzing the low-temperature WGS reaction up to 548 K while the o-Cu catalyst is inactive and with accumulated formate species on the surface (Figs. 2 and 3).

Therefore, our comprehensive experimental and theoretical calculation results clearly demonstrate a facet-dependent catalytic performance of Cu catalysts in low-temperature WGS reaction in which Cu{100} is the most active facet with the Cu–Cu suboxide interface as the active site. Interestingly, such a facet-dependent catalytic performance results mainly from a facet-dependent surface poisoning of the active site instead of from a facet-dependent intrinsic activity. All elementary surface reactions within the catalytic cycle can proceed smoothly at the Cu–$Cu_xO$ interface of active Cu{100} facets during low-temperature WGS reaction; however, the Cu–$Cu_xO$ interface of Cu{111} facets initially is also active, but will be soon self-poisoned by the accumulation of stable formate intermediate. These results nicely demonstrate a key feature of an active site of solid catalysts that an active site must be able to recycle during catalytic reactions.

**Design and realization of highly efficient ZnO/c-Cu catalyst.** The WGS reaction is mildly exothermic and thermodynamically favors low reaction temperatures. The above fundamental understandings of facet-dependent catalytic performance of Cu catalysts reveal a strategy of designing efficient copper-based catalysts for low-temperature WGS reaction using c-Cu NCs. Cu–ZnO-based catalysts are commercial catalysts for WGS reaction. A c-$Cu_2O$ NCs-supported ZnO catalyst with a ZnO weight ratio of 1% (1%wt-ZnO/c-$Cu_2O$) was thus synthesized and then reduced to acquire a c-Cu NCs-supported ZnO catalyst with a ZnO weight ratio of 1.67% (1.67%wt-ZnO/c-Cu) (Fig. 7a–d and Supplementary Fig. 24). Similarly, an o-$Cu_2O$ NCs-supported ZnO catalyst (1%wt-ZnO/o-$Cu_2O$) was also synthesized and then reduced to acquire an o-Cu NCs-supported ZnO catalyst

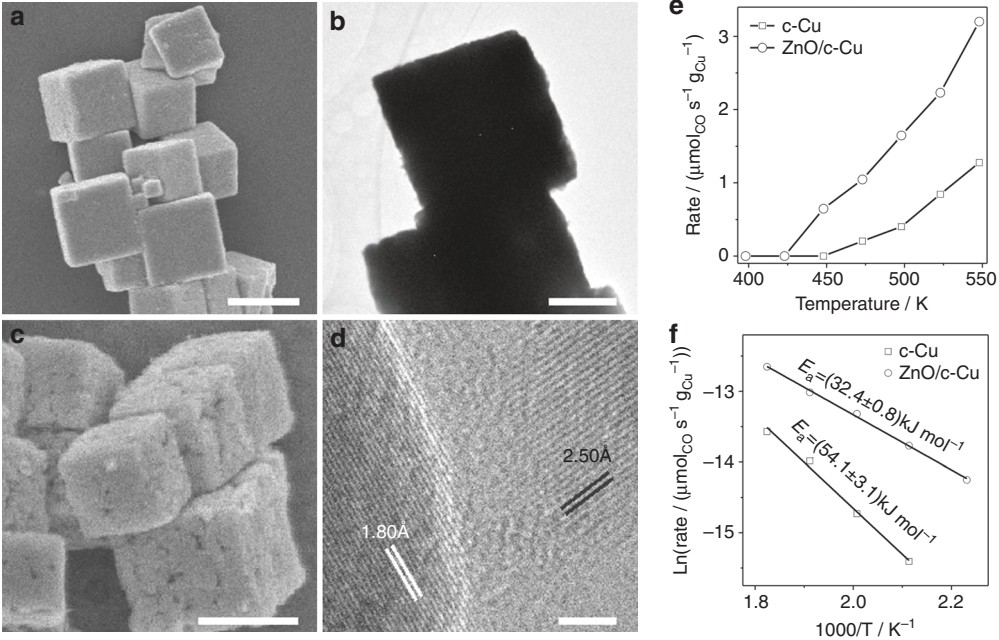

**Fig. 7** ZnO/c-Cu catalyst. The *scale bars* of **a**, **b** correspond to 1000 nm, that of **c** correspond to 500 nm, and that of **d** correspond to 2 nm. **a** Representative SEM images of 1%wt-ZnO/c-Cu$_2$O; **b** representative SEM, **c** TEM, **d** HRTEM images of 1.67%wt-ZnO/c-Cu catalysts; **e** Reaction rate ($\mu mol_{CO}$ s$^{-1}$ g$_{Cu}^{-1}$) of 1.67%wt-ZnO/c-Cu catalyst in the WGS reaction as a function of reaction temperature and **f** Arrhenius plot of 1.67%wt-ZnO/c-Cu in the WGS reaction. The data of c-Cu NCs are included for comparisons. The data of c-Cu NCs are included for comparisons. The lattice fringes of 1.80 and 2.50 Å, respectively, correspond to the spacing of Cu{200} (JCPDS card NO. 89-2838) and hexagonal ZnO{101} (JCPDS card NO 89-1397) crystal planes

(1.67%wt-ZnO/o-Cu) (Supplementary Figs. 24 and 25). It can be seen that both Cu(0) and Cu(I) exist on the surface of Cu NCs in ZnO/Cu catalysts. The ZnO/Cu NCs catalysts are much more active than the corresponding Cu NCs catalysts in catalyzing the WGS reaction (Fig. 7e and Supplementary Fig. 25e), and the calculated apparent activation energy of ZnO/c-Cu (32.4 ± 0.8 kJ mol$^{-1}$) (Fig. 7f) and ZnO/o-Cu (55.9 ± 3.9 kJ mol$^{-1}$) (Supplementary Fig. 25f) catalysts is much smaller than that of the corresponding Cu NCs catalysts. These results suggest that the copper-ZnO interface in ZnO/Cu NCs catalysts exhibits much higher intrinsic activity than the Cu–Cu$_x$O interface in Cu NCs catalysts. However, the detailed mechanism needs further study. Meanwhile, the ZnO/c-Cu catalyst is stable (Supplementary Fig. 26).

The apparent activation energy of ZnO/c-Cu catalyst is smaller than those of commercial Cu/ZnO/Al$_2$O$_3$ catalyst, Cu/ZnO(0001) (52 kJ mol$^{-1}$)[11], and ZnO/Cu (46 kJ mol$^{-1}$)[36] model catalysts, and is similar to those acquired on CeO$_x$/Cu(111) (30 kJ mol$^{-1}$)[37] and Cu/CeO$_2$(111) (37 kJ mol$^{-1}$)[11] model catalysts. It is noteworthy that the activity of CeO$_x$/Cu(111) model catalyst was evaluated between 573 and 673 K[37] under which the Cu (111) surface might restructure. Meanwhile, no Cu–CeO$_2$ and Cu–ZnO powder catalysts was reported to exhibit E$_a$ in WGS reaction as low as the CeO$_x$/Cu(111) and Cu/CeO$_2$(111) model catalysts, suggesting the presence of so-called materials gap. The ZnO/o-Cu catalyst exhibits a similar apparent activation energy to the commercial Cu/ZnO/Al$_2$O$_3$ catalyst but much larger than that of ZnO/c-Cu catalyst. Therefore, the active structure of ZnO/c-Cu catalyst should be intrinsically more active than those of ZnO/o-Cu and commercial Cu/ZnO/Al$_2$O$_3$ catalysts. This demonstrates a key role of the Cu structure in the Cu–ZnO-based catalysts in determining the catalytic activity in WGS reaction, in which the ZnO/c-Cu catalyst with the Cu{100} structures is highly efficient. Meanwhile, it could also be inferred that the copper structure of commercial

Cu/ZnO/Al$_2$O$_3$ catalyst should be dominated by the Cu{111} structures and its catalytic activity can be improved by engineering the Cu structure from the dominant Cu{111} structure into the Cu{100} structure. These findings demonstrate a successful experimental strategy of using catalyst NCs that realizes an all-chain investigation of heterogeneous catalysis from the fundamental understanding of active site and reaction mechanism to the structural design and realization of highly efficient catalysts.

## Methods

**Materials**. All chemicals were purchased from Sinopharm Chemical Reagent Co. Ltd. and used without further purification. Commercial Cu/ZnO/Al$_2$O$_3$ WGS catalyst was purchased from Alfa Aesar Chemical Co. Ltd. 5% CO/Ar, 0.432% CO/Ar, Ar (99.999%), CO (99.99%), C$_3$H$_6$ (99.95%), O$_2$ (99.999%), and N$_2$ (99.999%) were purchased from Nanjing Shangyuan Industrial Factory and used without further purification. ultrapure water (>18.5 MΩ) was used.

**Synthesis**. Synthesis of cubic and octahedral Cu$_2$O NCs followed the procedure reported by Zhang et al[38]. Typically, NaOH aqueous solution (2.0 mol L$^{-1}$; 10 mL) was added dropwise into CuCl$_2$ aqueous solution (0.01 mol L$^{-1}$; 100 mL) at 328 K (c-Cu$_2$O without poly(vinylpyrrolidone) (PVP), o-Cu$_2$O containing 4.44 g PVP ($M_w$ = 30 000)). After stirring for 0.5 h, ascorbic acid solution (0.6 mol L$^{-1}$; 10 mL) was added dropwise. The resulting solution was stirred at 328 K for 5 and 3 h to synthesize cubic and octahedra Cu$_2$O NCs, respectively. The resulting precipitate was collected by centrifugation, decanted by repeated washing with distilled water and absolute ethanol, and finally dried under vacuum at RT for 12 h. Synthesis of rhombic dodecahedral Cu$_2$O followed the procedure by Liang et al.[39]. Typically, 4 mL oleic acid (OA) and 20 mL absolute ethanol were added successively into 40 mL CuSO$_4$ aqueous solution (0.025 mol L$^{-1}$) under vigorous stirring. The solution was heated to 373 K and then 10 mL NaOH aqueous solution (0.8 mol L$^{-1}$) was added. After stirring for 5 min, 30 mL D-(+)-glucose aqueous solution (0.63 mol L$^{-1}$) was added. The resulting mixture was stirred at 373 K for additional 1 h and gradually turned into a brick-red color. The resulting precipitate was collected by centrifugation, decanted by repeated washing with distilled water and absolute ethanol, and finally dried in vacuum at room temperature (RT) for 12 h. The acquired cubic, octahedral, and rhombic dodecahedral Cu$_2$O NCs were denoted as c-Cu$_2$O, o-Cu$_2$O-PVP, and d-Cu$_2$O-OA, respectively.

Capping ligands on as-synthesized o-Cu$_2$O-PVP and d-Cu$_2$O-OA NCs were removed following a controlled oxidation procedure developed by Hua et al.[12].

Typically, $Cu_2O$ NCs (0.2 g) were placed in a U-shaped quartz microreactor and purged in the stream of $C_3H_6 + O_2 + N_2$ gas mixture ($C_3H_6:O_2:N_2$ = 2:1:22, flow rate: 50 mL min$^{-1}$) at room temperature for 0.5 h, and then heated to the desirable temperature (o-$Cu_2O$-PVP: 473 K; d-$Cu_2O$-OA: 488 K) at a rate of 5 K min$^{-1}$ and kept for 0.5 h, then the stream was switched to pure Ar (flow rate: 30 mL min$^{-1}$) in which the sample was cooled down to room temperature. The acquired octahedral and rhombic dodecahedral $Cu_2O$ NCs without capping ligands were denoted as o-$Cu_2O$ and d-$Cu_2O$, respectively.

Cu NCs with various morphologies were synthesized via a morphology-preserved reduction method. $Cu_2O$ NCs (0.1 g) were placed in a quartz microreactor and purged in the stream of 5% CO balanced with Ar (flow rate: 30 mL min$^{-1}$) at room temperature for 0.5 h, and then heated to 548 K at a rate of 1 K min$^{-1}$ and kept for 2 h, then cooled down to room temperature. The acquired cubic, octahedral and rhombic dodecahedral Cu NCs were denoted as c-Cu, o-Cu, and d-Cu, respectively.

$Cu_2O$ NCs-supported ZnO catalysts (ZnO/$Cu_2O$) were synthesized by incipient wetness impregnation method. Typically, $Cu_2O$ NCs (0.2 g) were adequately incipient wetness impregnated with 200 µL aqueous solution containing calculated amounts of zinc nitrate ($Zn(NO_3)_2·6H_2O$), and the resulting mixture was dried under vacuum at room temperature overnight and then heated at 623 K for 2 h in high-pure Ar with a flow of 30 mL min$^{-1}$. The acquired ZnO/$Cu_2O$ catalysts were then reduced in 5% CO/Ar with a flow of 30 mL min$^{-1}$ to 423 K at a rate of 1 K min$^{-1}$ and kept for 2 h to prepare corresponding ZnO/Cu catalysts that were stored in vacuum oven.

Details on structural characterizations, catalytic activity evaluation, and DFT calculations can be found in the Supplementary Methods.

**Data availability**. All data are available from the authors on reasonable request.

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

## Acknowledgements

This work was financially supported by National Basic Research Program of Ministry of Science and Technology of China (2013CB933104, 2013CB834603), National Natural Science Foundation of China (21525313, 21173204, U1332113, 91645202), Fundamental Research Funds for the Central Universities of Ministry of Education of China (WK2060030017), Hefei Science Center of Chinese Academy of Sciences

(2015HSC-UP014), and Collaborative Innovation Center of Suzhou Nano Science and Technology.

## Author contributions

W.H. designed and supervised the project. W.-X.L. supervised the DFT calculations. Z.Z. carried out all experiments. S.-S.W. carried out all DFT calculations. R.S., T.C., L.L., Y.G., X.C., and J.L. assisted with the experiments. W.H. and Z.Z. analyzed all the results. S.-S.W. and W.-X.L. analyzed the DFT calculation results. W.H., Z.Z., S.-S.W. and W.-X.L. prepared the manuscript and other authors commented on the manuscript.

## Additional information

**Competing interests:** The authors declare no competing financial interests.

