## [Peer Review file · Nature Communications]

Reviewers' comments:

Reviewer #1 (Remarks to the Author):

This is a detailed study presenting the use of well defined, ligand free, metallic and oxidized copper nanocrystals to determine the most active phase for the water-gas-shift (WGS) reaction. The authors convincingly show that open surfaces of copper are more reactive than the thermodynamically preferred Cu(111) phase, but more importantly that the presence of Cu₂O islands helps the metal nanocrystals to dissociate water, the rate limiting step on metal surfaces for the WGS reaction. The authors also show that by depositing a small coverage of ZnO on top of the Cu nanocrystals they obtain a catalyst that is even more active than the commercially used Cu/ZnO/Al₂O₃ catalyst, due to the formation of active bifunctional metal/oxide interphases. I find the paper worth of publishing, but there are a few comments that the authors should address before publication.

1-The authors assign the IR features at 2107 and 2109 cm⁻¹ in Figure 1.F to adsorption of CO on Cu(I) sites. This has been done in the literature on powder samples, but on well-defined single crystals it has been shown that these could be better assigned to under-coordinated Cu(0) sites, and that Cu(I) sites lead to CO adsorption with >2,110 cm⁻¹ features (see J. Phys. Chem. C 2014, 118, 15902–15909 and references therein). Since Cu₂O can be very easily reduced to Cu at room temperature, the top layer of a Cu₂O NC can consist of metallic Cu sites after a short exposure to CO. In the same Figure 1.F, it is very unlikely that the features at 2193 and 2225 cm⁻¹ are associated to CO₂ as assigned in the manuscript. Linear CO₂ interacts very weakly with any surface, and even when it does it would lead to IR features close to gas phase CO₂ (~2,350 cm⁻¹) as it does when it is trapped in zeolites at room temperature.

2-The authors referred to carbonate species (CO₃²⁻), while more likely these are carboxylate (CO₂⁻) species. Due to the similarity in the XPS features, the assignments of carbonate, formate and carboxylate species in Figure S5 should be correlated to the corresponding IR data (similar to Figure 3.D) in order to obtain a more convincing set of assignments.

3-The authors use of "quasi in situ XPS" is not correct. In situ means "in place" of the reaction, like they do for their DRIFT measurements. They should just mention that they measure XPS on their samples after reactions without exposure to air.

4-In the figures describing the DFT calculations it is almost impossible to see the configuration of the intermediates. Is the COOH species they mention in the text a carboxyl species with the carbon on the metal or what/where? The authors should add bigger figures with the critical intermediates (such as the carboxyl COOH species) in order to clarify what and where are these adsorbates.

Minor:

Page 4: "Cu₂O NCs are well reserved.." should be "Cu₂O NCs are well preserved..". There are several other instances where the term reserved is used and the meaning of it is not clear.

Page 4: the paragraph where the sentence "..., and reasonably are smaller than the corresponding .." is used is not clear and should be re-written.

Page 5: same with "..was while no feature was observed on.."

Reviewer #2 (Remarks to the Author):

Review of "The Active Cu Structure for Low-Temperature Water Gas Shift Reaction"

Summary:

This paper describes the reactivity of Cu nanocrystals (NCs) with distinct exposed facets towards the water-gas-shift reaction. The NCs were synthesized as Cu_2O , and were characterized and probed both before and after reduction. The authors provide evidence that the (100) facet of Cu is very active. This kind of work is important to merge the fields of single crystal surface science and catalysis engineering. However, the authors have not fully characterized their catalysts to the level that Nat. Commun. would require and have made claims that over-step what the data allows. The kinetic test analysis has flaws as does the computational part. For these reasons I do not recommend acceptance of the manuscript in the present form. My concerns are listed below in more detail:

Major Concerns:

- 1) It is well known that Cu_2O is relatively unstable. The Auger peaks for Cu(I) and Cu(II) come at nearly identical kinetic energies, do the authors have any evidence that CuO isn't also present on the surface of these nanoparticles?
- 2) On p. 6 the authors compare the activity of the NCs to Cu/ZnO/ Al_2O_3 . But it is not an accurate comparison to calculate catalytic activity based on total Cu atoms for the Cu/ZnO/ Al_2O_3 and only surface Cu atoms for the NCs. The activity calculation needs to be the same for all catalysts being compared. The commercial catalyst is several orders of magnitude more active than the NCs. Reaction rates (per g catalyst or per SA) may be used for comparison, but not TOFs.
- 3) On p. 7 the authors discuss that both the Cu and Cu_2O NCs are equivalent in activity under reaction conditions. The authors then state this a evidence for Cu(0) as the active site, as opposed to Cu(I). However, this statement is inaccurate and should be removed, as the authors go on to show how a partially oxidized surface is much more active than the bare metal. The data seems to suggest that the catalyst is very dynamic and doesn't exist in purely one oxidation state, rather a mix of the oxidation states is likely created.
- 4) Related to the previous point, the authors should characterize their post-reaction NCs to quantify how the catalyst has changed, which would shed light on the active state of the catalyst.
- 5) The poisoning by formate mechanism is surprising at first because formate should be much more weakly bound than the other listed carbon species. What seems to lead to the poisoning is that the formate is made at the active site and blocks it. Is it possible for the other carbon species to be removed in order to more definitively identify the formate species as the poisoning adsorbate?
- 6) On p. 9 the authors state that the CO oxidation takes place on the Cu - Cu suboxide interface. However the data only shows this oxygen species as

being active and there is no evidence for the reaction taking place exclusively at the interface, as opposed to on top of the suboxide. It should be made clearer to the reader that the data presented does not identify if the reaction takes place at the interface or on top of the suboxide regions.

- 7) The interface as the locus of activity for the inverse structures of ZnO/Cu single crystal surfaces has been demonstrated by the work of Rodriguez et al. (not included in the refs. here; ref. 27 reaches the same conclusion for ceria). At the interface, the oxidation state of the active Cu species has been debated in the literature, and convergence to Cu(I) is shown. In the work by Flytzani-Stephanopoulos et al., the WGS reaction has been found to occur on cationic metal species, including Au(I), Pt(II), Cu(I), etc. In other words, the reduction to the metal is not complete. See for example, Rui Si et al., "Structure sensitivity of the low-temperature water-gas shift reaction on Cu-CeO₂ catalysts", *Catal. Today*, **2012**, 180, 68. How to probe the interface with any spectroscopy in the preponderance of metallic Cu as in the authors' NCs is a formidable and perhaps untenable task.
- 8) A cautionary statement in the kinetics work. Have the authors conducted their tests at steady-state and from high-T to low-T? if not, structural changes at the high temperatures, could have caused lower rates, i.e. an erroneous apparent activation energy, which can explain the values of ~ 30 kJ/mol reported for the Cu/ZnO catalysts. There is no reason for the activation energy to be different, if the same cuprous species is the active site in all such catalysts, i.e. the mechanism does not change. Cu should not be compared to other metals like gold. The latter has its own "signature" with Ea lower than that of Cu (or Pt) on any support.
- 9) The oxidation of flat copper surfaces has been extensively studied. Although the exact structure of Cu under reaction conditions is not known, the design of the DFT model for the Cu-Cu₂O interface should resemble some of the motifs found on carefully studied single crystal surfaces (Besenbacher and Norskov, *Prog. Surf. Sci.* **1993**, 44, 5). For example, O on Cu(111) readily reconstructs into ring structures over a wide range of O coverages and temperatures (Matsumoto et al. *Surf. Sci.*, **2001**, 471, 225; Soon et al. *Phys. Rev. B*, **2006**, 73, 165424; Yang et al. *J. Phys. Chem. C*, **2010**, 114, 17042), but a structure like the zig-zag pattern that the authors chose as their model has never been observed experimentally. This is unfortunate given the large library of data on partially oxidized Cu surfaces, and dramatically diminishes the impact of the calculations with the structure chosen. The authors should provide some justification to the reader for the design of their DFT model, and should also provide the reader with information of observed O/Cu structures from the literature, including some of the references listed above, and how their structures differ.

Minor Concerns:

- 10) Recently there has been a study on Cu single crystals from the Salmeron group showing that the (111) facet has the best activity towards CO oxidation. (Eren et al. *J. Phys. Chem. C* **2015**, *119*, 14669) CO oxidation is a crucial step in the water-gas-shift reaction; could the authors comment on the inverse in the activity hierarchy for these two reactions?
- 11) I suggest a modification to the title from “the active Cu structure...” to “the most active Cu facet...” because all Cu structures investigated are indeed active, this work is aimed towards finding the most active facets (as apposed to general structure such as nanoparticle size or shape).
- 12) The authors may want to consider re-labeling of the “suboxide” species, because it suggests that the oxide is subsurface. However the authors find that there is O on the surface at these suboxide regions, so perhaps calling these sites simply “oxide” would be more fitting.

Reviewer #3 (Remarks to the Author):

This manuscript describes a promising approach to determine the active site structure of Cu-based catalysts for the low temperature water-gas shift (WGS) reaction. The information is then used to prepare an improved catalyst sample. The WGS reaction is of great industrial importance and the currently used catalyst, Cu/ZnO/Al₂O₃ is also Cu-based; thus, this topic is of high relevance and could be appealing to a large audience.

Despite the potential impact of the work, there are several gaps in the argumentation. These gaps are described below and require major revisions before this manuscript can be reviewed again.

The interpretation of the TPRS data on page 7 leads the authors to claim that H₂ desorption is the rate-limiting step of Cu-catalyzed WGS. I don't see sufficient evidence for this conclusion. If the active site is the Cu/Cu-suboxide interface, CO₂ desorption could simply be a result of Cu-suboxide reduction by CO. I assume similar CO₂ desorption peaks would be observed in the absence of H₂O. Also, if H₂ desorption was rate-limiting, why did the authors not simulate this step with DFT? The adsorption/desorption of H₂ in Table S5 is assumed to be quasi-equilibrated and no barrier is reported.

Ultimately, the authors attribute WGS reactivity to the presence of Cu/Cu-suboxide interfaces. With this active site assignment, many of the performance data are questionable. The BET surface area would not be a good way to normalize the TOF. Instead, the authors should normalize by the number of interface sites. The observed differences may simply be a difference in how well the different Cu facets oxidize to a suboxide, or how well the epitaxial relationship between the Cu metal and the suboxide matches. It is likely that the thermodynamically most stable (111) facet is not easily oxidized. From the DFT results, the authors could easily extract the interfacial energy between Cu(111) or Cu(100) and Cu₂O, and determine if there is a significant stability difference.

Side note: a brief clarification why the BET surface area of larger o-Cu NCs (937 nm) is larger than that of smaller c-Cu NCs (877 nm) would be appropriate. It is likely due to the higher site density of the (111) planes.

The high surface roughness is a concern. A rough surface will necessarily have defect and step sites. Were there any efforts to quantify the amount of surface defects? The CO vibration experiments should exhibit signals of CO molecules strongly adsorbed on steps/defects. Did the authors try to compare the signal intensities to get a rough estimate of the ratio of step to terrace sites?

The DFT results suggest that there is no significant difference between Cu(111) and Cu(100) model interfaces with Cu₂O present. The only difference was speculated to be the formate coverage based on the formation and decomposition barriers. Such speculation is rather unreliable and often fails, even in simple cases. The coverage of formate depends on the coverage of H and CO₂, which in turn depend on everything else. Also, if formate is a spectator species, it will be formed until it reached its saturation coverage. At this point the formation and decomposition barriers are no longer relevant to estimate the formate coverage at the interface under reaction conditions. Hence, I see value in the DFT work to assign the active site to the Cu/Cu₂O interface, but I do not see agreement with the experimentally observed activity increase on the (100) facet.

Were the transition states found in the NEB method verified by vibrational analysis showing a single imaginary mode? I would be beneficial if the authors clearly stated that zero point energies and entropy corrections were neglected.

Could the authors please clarify if the reaction energies and barriers are taken with respect to isolated reactants/products? The equation in the SI suggests this but an explicit statement is preferred. If energies are reported at infinite separation, could the authors also include the

interaction energies for the co-adsorbed states that form the initial and final states of their NEB calculations?

The manuscript must be edited for grammar and spelling. Most importantly "morphology -reserved" should probably be "morphology-preserved".

Authors' Reply to Reviewer 1's Comments and Revisions

Comment: This is a detailed study presenting the use of well defined, ligand free, metallic and oxidized copper nanocrystals to determine the most active phase for the water-gas-shift (WGS) reaction. The authors convincingly show that open surfaces of copper are more reactive than the thermodynamically preferred Cu(111) phase, but more importantly that the presence of Cu₂O islands helps the metal nanocrystals to dissociate water, the rate limiting step on metal surfaces for the WGS reaction. The authors also show that by depositing a small coverage of ZnO on top of the Cu nanocrystals they obtain a catalyst that is even more active than the commercially used Cu/ZnO/Al₂O₃ catalyst, due to the formation of active bifunctional metal/oxide interphases. I find the paper worth of publishing, but there are a few comments that the authors should address before publication.

Author Reply: We appreciate the reviewer's positive and valuable comments very much. We have seriously considered the reviewer's comments and revised the manuscript accordingly. We hope that the revised manuscript will be suitable for the publication in *Nature Communications*.

Comment 1: The authors assign the IR features at 2107 and 2109 cm⁻¹ in Figure 1.F to adsorption of CO on Cu(I) sites. This has been done in the literature on powder samples, but on well-defined single crystals it has been shown that these could be better assigned to under-coordinated Cu(0) sites, and that Cu(I) sites lead to CO adsorption with >2,110 cm⁻¹ features (see *J. Phys. Chem. C* 2014, 118, 15902–15909 and references therein). Since Cu₂O can be very easily reduced to Cu at room temperature, the top layer of a Cu₂O NC can consist of metallic Cu sites after a short exposure to CO. In the same Figure 1.F, it is very unlikely that the features at 2193 and 2225 cm⁻¹ are associated to CO₂ as assigned in the manuscript. Linear CO₂ interacts very weakly with any surface, and even when it does it would lead to IR features close to gas phase CO₂ (~2,350 cm⁻¹) as it does when it is trapped in zeolites at room temperature.

Author Reply: We appreciate the reviewer's insightful comments very much and we have seriously considered these comments. We agree with that caution must be taken to assign the copper species probed with CO adsorption due to the likely surface reduction induced of copper oxides by CO. All previously observed surface reductions occur at certain temperatures, but during our investigations CO adsorption were carried out at 123 K, as described in the methods section of the supporting information, where the surface reduction of Cu₂O NCs does not likely occur. Meanwhile, the vibrational feature of CO adsorbed at Cu(I) sites were reported to vary with the Cu(I) structure. Although mostly it locates above 2110 cm⁻¹ at supported and well-dispersed Cu(I), it can locate below 2110 cm⁻¹ (for examples, Martínez-Arias, A., Fernández-García, M., Soria, J., Conesa, J. C. Spectroscopic study of a Cu/CeO₂ catalyst subjected to redox treatments in carbon monoxide and oxygen. *J. Catal.* **182**, 367-377 (1999) & Xu, F., Mudiyansele, K., Baber, A. E., Soldemo, M., Weissenrieder, J., White, M. G., Stacchiola, D. J. Redox-mediated reconstruction of copper during carbon monoxide oxidation. *J. Phys. Chem. C* **118**, 15902-15909 (2014)). It can be also seen from our results that CO adsorbed at Cu surface exhibits vibrational features below 2100 cm⁻¹. Therefore, we believe that it is reasonable to assign

the IR features at 2107 and 2109 cm^{-1} in Figure 1.F to CO adsorbed at Cu(I) sites of Cu_2O NCs. After CO adsorption on Cu_2O octahedra at 123 K, vibrational bands at ~ 2193 and 2225 cm^{-1} were also observed on o- Cu_2O in addition to the vibrational features of adsorbed CO. We have studied this carefully in our previous publications (Ref. 12 and 19 in the revised manuscript), in which, by comparing with the IR spectra of CO_2 adsorption on Cu_2O NCs, both features were assigned to CO_2 adsorbed at the coordination-unsaturated Cu(I) sites of o- Cu_2O . The adsorbed CO_2 is formed by the reaction of CO with adsorbed oxygen species on o- Cu_2O introduced during the controlled oxidation process of as-synthesized PVP-capped o- Cu_2O NCs to remove the capping ligand. Again, due to the very low adsorption temperature, the formed CO_2 can stably adsorb at the coordination-unsaturated Cu(I) sites of o- Cu_2O . In the present manuscript, we cite our previous results on Cu_2O NCs to briefly explain the experimental observations.

In reply to the reviewer, in the revised manuscript, we have added the CO adsorption temperature (123 K) in both the text and the Figure 1 caption (Please see Line 6 on Page 6 and Figure 1 caption on Page 25); we have also included both papers of *J. Catal.* **182**, 367-377 (1999) and *J. Phys. Chem. C* **118**, 15902-15909 (2014) as Ref. 20 and 21 to further support our assignment; we have also re-written the relevant sentences as the following to clarify the results and discussion: **“Figure 1F compares in-situ DRIFTS spectra of CO adsorption at 123 K on Cu_2O and Cu NCs. Agreeing with previous reports,^{12,19-21} a vibrational band was observed at $\sim 2108 \text{ cm}^{-1}$ on o- Cu_2O and d- Cu_2O and assigned to CO adsorbed at the Cu(I) sites exposed on the {111} and {110} facets while no feature was observed on the Cu_2O cubes enclosed with the O-terminated Cu_2O (100) facets; and vibrational bands at ~ 2193 and 2225 cm^{-1} were also observed on o- Cu_2O and assigned to CO_2 adsorbed at the coordination-unsaturated Cu(I) sites formed by the reaction of CO with adsorbed oxygen species^{12,19}.”** (Please see Lines 6-12 on Page 6); we have also re-ordered the references accordingly.

Comment 2: The authors referred to carbonite species (CO_2^-), while more likely this are carboxylate (CO_2^-) species. Due to the similarity in the XPS features, the assignments of carbonate, formate and carboxylate species in Figure S5 should be correlated to the corresponding IR data (similar to Figure 3.D) in order to obtain a more convincing set of assignments.

Author Reply: We appreciate the reviewer’s insightful comments very much and we have seriously considered these comments. After carefully looking into the literatures and comparing the differences between the carbonite and carboxylate species, we agree with the reviewer that the carbonite species in our original manuscript should be re-assigned to the carboxylate species. Meanwhile, we have also followed the reviewer’s suggestion to examine the in-situ DRIFTS spectra of as-synthesized Cu NCs with the spectra of corresponding Cu_2O NCs as the reference spectra (shown below). The spectra are dominated by the significant attenuation of the Cu_2O vibrational features (652 , 794 and 1123 cm^{-1}), but the formation of carboxylate (1277 cm^{-1}) and carbonate (1235 , 1455 and 1506 cm^{-1}) species could be observed on all Cu NCs and formate (1588 cm^{-1}) species could be observed exclusively on o-Cu NCs. These results are in consistence with the XPS results.

In reply to the reviewer, we have revised the “carbonite” to the “carboxylate” throughout all the revised manuscript; we have added the in-situ DRIFTS spectra of as-synthesized Cu NCs as Figure S6 in the revised supporting information (Please see Page S18 in the supporting information) and described and discussed the relevant results as the following: “In the corresponding O 1s and C 1s XPS spectra (Figure S5), all Cu NCs exhibit three O 1s components at 529.5, 530.3 and 531.8 eV and three C 1s components at 284.8, 289.0 and 287.7 eV while o-Cu NCs exhibit an additional minor C 1s component at 286.6 eV. The O 1s components at 529.5 and 531.8 eV could be respectively assigned to oxygen adatoms and hydroxyl groups/oxygenates species while the O 1s component at 530.3 eV and the Cu(I) feature were observed to vary simultaneously for all acquired Cu NCs and could be assigned to Cu suboxide (Cu_xO , $x \geq 10$).^{13, 14} The C 1s components at 284.8, 289.0, 287.7 and 286.6 eV could be assigned respectively to adventitious carbon, carbonate, carboxylate and formate species.¹⁵ The in-situ DRIFTS spectra of the reduction processes of Cu_2O NCs into Cu NCs (Figure S6) show the significant attenuation of the Cu_2O vibrational features (652 , 794 and 1123 cm^{-1})¹⁶ and the formation of carboxylate (1277 cm^{-1}) and carbonate (1235 , 1455 and 1506 cm^{-1}) species on all Cu NCs and formate (1588 cm^{-1})^{17, 18} species exclusively on o-Cu NCs, in consistence with the XPS results. All the observed surface species on the acquired Cu NCs are common on metal powder catalysts.” (Please see the texts from Line 14 on Page 5 to Line 5 on Page 6). We have also re-ordered the figures and the references accordingly.

Comment 3: The authors use of “quasi in situ XPS” is not correct. In situ means “in place” of the reaction, like they do for their DRIFT measurements. They should just mention that they measure XPS on their samples after reactions without exposure to air.

Author Reply: We appreciate the reviewer’s critical comments and kind suggestions very much and we have seriously considered them. During our experiments, two types of XPS measurements were carried: one was routine measurements; the other was measurements of the samples subjected to various treatments without exposure to air, which we defined as “quasi in situ XPS” in order to distinguish from the routine XPS measurements. However, we agree with the reviewer that XPS measurements of the samples subjected to various treatments without exposure to air are not appropriate to be defined as “quasi in situ XPS”.

In reply to the reviewer, we have removed the term “quasi in situ XPS” throughout all the revised manuscript and mentioned it as “XPS without exposure to air” in the contexts needed.

Comment 4: In the figures describing the DFT calculations is almost impossible to see the configuration of the intermediates. Is the COOH species they mention the text a carboxyl species with the carbon on the metal or what/where? The authors should add bigger figures with the critical intermediates (such as the carboxyl COOH species) in order to clarify what and where are these adsorbates.

Author Reply: We appreciate the reviewer’s critical comments and kind suggestions very much and we have seriously considered them. We accept the reviewer’s suggestions and have split the original Figure 4 into Figure 5 and Figure 6 (shown below) in the revised manuscript (Please see Figure 5 on Page 33 and Figure 6 on Page 34) and included an additional Figure S18 with the structures of all important adsorbates and intermediates (shown below) in the revised supporting information manuscript (Please see Figure S18 on Page S31) in order to provide high-quality figures. We have also made revisions in the texts and re-ordered the figures accordingly.

Figure 5

Figure 6

(A) $\text{Cu}_2\text{O-Cu(111)}$

(B) $\text{Cu}_2\text{O-Cu(100)}$

Figure S18

Minor comments: Page 4: “Cu₂O NCs are well reserved..” should be “Cu₂O NCs are well preserved..”. There are several other instances where the term reserved is used and the meaning of it is not clear.

Author Reply: We appreciate the reviewer’s careful reading and kind suggestions very much and have replaced “reserved” with “preserved” throughout the revised manuscript and supporting information.

Minor comments: Page 4: the paragraph where the sentence “.., and reasonably are smaller than the corresponding ..” is used is not clear and should be re-written.

Author Reply: We appreciate the reviewer’s careful readings very much and have rewritten the sentence as the following in the revised manuscript: **“However, the acquired Cu NCs exhibit rough surfaces likely resulting from the lattice mismatch between Cu₂O and Cu. The sizes of acquired Cu NCs are smaller than those of corresponding Cu₂O NCs.”** (Please see Lines 21-23 on Page 4).

Minor comments: Page 5: same with “..was while no feature was observed on..”

Author Reply: We appreciate the reviewer’s careful readings very much and have rewritten the sentence as the following in the revised manuscript: **“Agreeing with previous reports,^{12,19-21} a vibrational band was observed at ~2108 cm⁻¹ on o-Cu₂O and d-Cu₂O and assigned to CO adsorbed at the Cu(I) sites exposed on the {111} and {110} facets while no feature was observed on the Cu₂O cubes enclosed with the O-terminated Cu₂O (100) facets;”** (Please see Lines 7-10 on Page 12).

Authors' Reply to Reviewer 2's Comments and Revisions

Summary: This paper describes the reactivity of Cu nanocrystals (NCs) with distinct exposed facets towards the water-gas-shift reaction. The NCs were synthesized as Cu₂O, and were characterized and probed both before and after reduction. The authors provide evidence that the (100) facet of Cu is very active. This kind of work is important to merge the fields of single crystal surface science and catalysis engineering. However, the authors have not fully characterized their catalysts to the level that Nat. Commun. would require and have made claims that over-step what the data allows. The kinetic test analysis has flaws as does the computational part. For these reasons I do not recommend acceptance of the manuscript in the present form. My concerns are listed below in more detail.

Author Reply: We appreciate the reviewer's positive and valuable comments very much. We have seriously considered the reviewer's comments and revised the manuscript accordingly. We hope that the revised manuscript will be suitable for the publication in *Nature Communications*.

Comment 1) It is well known that Cu₂O is relatively unstable. The Auger peaks for Cu(I) and Cu(II) come at nearly identical kinetic energies, do the authors have any evidence that CuO isn't also present on the surface of these nanoparticles?

Author Reply: We appreciate the reviewer's insightful comments very much and we have seriously considered them. Generally, Cu(0) and Cu(I) exhibit the very similar Cu 2p binding energies but different Cu LMM AES kinetic energies, and Cu(I) and Cu(II) exhibit the very similar Cu LMM AES kinetic energies but different Cu 2p binding energies. Thus a combination of Cu 2p XPS and Cu LMM AES spectra can distinguish among Cu(0), Cu(I) and Cu(II) species. We have measured Cu 2p XPS and LMM AES spectra for all our samples. The Cu 2p XPS spectra demonstrate the absence of Cu(II) feature on Cu₂O and Cu NCs. The stability of Cu₂O NCs is related to their quite large sizes.

In reply to the reviewer, we have included Cu 2p XPS spectra for all Cu₂O and Cu NCs in the revised supporting information (shown below) (Please see Figure S2 on Page S13, Figure S5 on Page S17, Figure S15 on Page S27, and Figure S16 on Page S28). We have also made revisions in the texts and re-ordered the figures accordingly.

Figure S2

Figure S5

Figure S15

Figure S16

Comment 2) On p. 6 the authors compare the activity of the NCs to Cu/ZnO/Al₂O₃. But it is not an accurate comparison to calculate catalytic activity based on total Cu atoms for the Cu/ZnO/Al₂O₃ and only surface Cu atoms for the NCs. The activity calculation needs to be the same for all catalysts being compared. The commercial catalyst is several orders of magnitude more active than the NCs. Reaction rates (per g catalyst or per SA) may be used for comparison, but not TOFs.

Author Reply: We appreciate the reviewer's insightful comments very much and we have seriously considered them. It is very difficult to accurately compare the reaction rates of un-supported Cu NCs and supported Cu/ZnO/Al₂O₃ catalyst due to the difficulty of identifying the surface Cu atom number of Cu/ZnO/Al₂O₃ and even unlikely task of identifying the active site number. Thus in the original manuscript we referred to the papers comparing the reaction rates of Cu single crystals and Cu/ZnO/Al₂O₃ (J. A. Rodriguez et al., Science 2007, 318 1757-1760; J. A. Rodriguez et al., Angew. Chem. Int . Ed. 2007, 46, 1329-1332.) and adopted their methods. But we definitely agree with the reviewer that such comparisons are not appropriate and accept the reviewer's suggestion to revise.

In reply to the reviewer, we have used the surface Cu atom-specific reaction rates, instead of TOF, to describe the catalytic activity of Cu NCs in the revised manuscript (Please see Figure 2 on Page 30) and the mass-specific reaction rates of to describe the catalytic activity of Cu/ZnO/Al₂O₃ in the revised supporting information (Please see Figure S8 on Page S20). We have also removed the data of Cu/ZnO/Al₂O₃ from the revised Figure 2A and compared the intrinsic catalytic activity of Cu NCs and Cu/ZnO/Al₂O₃ with the calculated apparent activation energies (Figure S2C). We have revised the relevant texts and discussion in the revised manuscript as the following: **“Various types of Cu NCs exhibit morphology-dependent catalytic activity in the WGS reaction. CO conversion over the c-Cu and d-Cu NCs was observed respectively at 473 and 498 K but not for the o-Cu NCs up to 548 K (Figure S7). Catalytic activity of commercial Cu/ZnO/Al₂O₃ catalyst in the WGS reaction was also evaluated (Figure S8). Calculated on the basis of surface Cu atoms derived from the specific surface areas of c-Cu and d-Cu NCs and the Cu atom densities of Cu (100) and (110) surfaces, the surface Cu atom-specific reaction rates of c-Cu NCs are much higher than those of d-Cu NCs (Figure 2A). The c-Cu and d-Cu NCs are also catalytically stable at**

548 K (Figure 2B and Figure S7), and the SEM images (Figure S9) demonstrate that the Cu NCs preserve their morphologies after the activity evaluation up to 548 K. Results of a Weisz–Prater analysis and a Mears analysis²⁶ in Supplemental Information show the absence of mass and heat transfer limitations under our reaction conditions. Thus the Arrhenius plots (Figure 2C) were made to calculate the apparent activation energy (E_a) of the WGS reaction catalyzed by the c-Cu, d-Cu and Cu/ZnO/Al₂O₃ respectively as 54.1±3.1, 68.4±8.0 and 51.6±3.7 kJ·mol⁻¹. Therefore, employing various types of Cu NCs, we demonstrate the obvious facet effect of Cu particles in the low-temperature WGS reaction with the Cu{100} facet as the most active facet.” (Please see the last Paragraph on Page 7).

Comment 3) On p. 7 the authors discuss that both the Cu and Cu₂O NCs are equivalent in activity under reaction conditions. The authors then state this a evidence for Cu(0) as the active site, as opposed to Cu(I). However, this statement is inaccurate and should be removed, as the authors go on to show how a partially oxidized surface is much more active than the bare metal. The data seems to suggest that the catalyst is very dynamic and doesn't exist in purely one oxidation state, rather a mix of the oxidation states is likely created.

Author Reply: We appreciate the reviewer's careful readings and insightful comments very much and we have seriously considered them. In the original manuscript we wanted to express that Cu, instead of Cu₂O, as the active copper phase in the WGS reaction. Such an argument is reasonable since Cu(I) is present on the metallic Cu surface.

In reply to the reviewer, we have re-written the sentence as the following in the revised manuscript: **“These results demonstrate Cu, instead of Cu₂O, as the active copper phase in the WGS reaction and further support the Cu{100} facet as the most active facet.”** (Please see the last sentence on Page 8).

Comment 4) Related to the previous point, the authors should characterize their postreaction NCs to quantify how the catalyst has changed, which would shed light on the active state of the catalyst.

Author Reply: We appreciate the reviewer's insightful comments very much and we have seriously considered them. In the supporting information we have already provided XRD patterns (Figure S12) and SEM images (Figure S13) for supporting our argument that Cu₂O NCs undergo the in-situ morphology-preserved reduction into the corresponding Cu NCs during the WGS reaction up to 548 K. We have measured the Cu LMM AES spectra of Cu₂O NCs and Cu NCs mixed with Al₂O₃ after the catalytic activity evaluations in the WGS reaction up to 548 K (shown below). The results show that the Cu LMM AES spectra of Cu₂O NCs and Cu NCs mixed with Al₂O₃ after the catalytic activity evaluations in the WGS reaction up to 548 K are very similar. However, at that time, we did not have access to the XPS facility capable of treating the catalysts and measuring XPS without exposure to air, thus the spectra were measured with the catalysts taken out from the reactor and exposed to air, resulting in that the Cu(I)/Cu(0) ratios on the used Cu NCs surfaces are higher than those measured without exposure to air (Figure S15) due to the occurrence of the surface oxidation. But all these structural characterization results adequately prove that the Cu₂O NCs undergo the in-situ morphology-preserved reduction into the

corresponding Cu NCs in the view of both bulk structure and surface structure.

In reply to the reviewer, we have added the Cu LMM AES spectra of Cu₂O NCs and Cu NCs mixed with Al₂O₃ after the catalytic activity evaluations in the WGS reaction up to 548 K as Figure S14 on Page S26 in the revised supporting information to demonstrate their surface compositions and explained the results in the figure caption. We have rewritten the relevant texts in the revised manuscript as the following: **“We found that, when directly used as the catalysts for the WGS reaction, various types of Cu₂O NCs exhibited the steady-state catalytic activity and calculated apparent activation energy almost identical to those of the corresponding Cu NCs (Figure S11). The structural characterization results (Figures S12-S14) demonstrate that the Cu₂O NCs undergo the in-situ morphology-preserved reduction into the corresponding Cu NCs with the presence of both Cu(0) and Cu(I) on their surfaces during the WGS reaction up to 548 K.”** (Please see Lines 16-22 on Page 8).

Comment 5) The poisoning by formate mechanism is surprising at first because formate should be much more weakly bound than the other listed carbon species. What seems to lead to the poisoning is that the formate is made at the active site and blocks it. Is it possible for the other carbon species to be removed in order to more definitively identify the formate species as the poisoning adsorbate?

Author Reply: We appreciate the reviewer’s insightful comments very much and we have seriously considered them. As described and discussed in the Section Reaction mechanism, our comprehensive characterization results shown in Figure 3 and Figure S15 clearly demonstrate the accumulation of formate and hydroxyl intermediates during the WGS reaction up to 548 K on the o-Cu surface but not on the c-Cu and d-Cu surfaces. In connection to the observations that c-Cu and d-Cu are catalytic active while o-Cu is poisoned by a stable hydrogen-containing intermediate on the surface, it is reasonable to conclude that the accumulating formate and hydroxyl intermediates should lead to the self-poisoning and subsequent inactivity of o-Cu NCs. However, it is noteworthy that the accumulation of formate and hydroxyl intermediates on the o-Cu surface can be alleviated at higher reaction temperatures, as the o-Cu NCs were observed to

become active in catalyzing the WGS reaction at 623 K and above (shown below). But the catalysts seriously restructure at such high reaction temperatures, thus we did not show these results in the original manuscript. However, we have repeatedly emphasized in the manuscript that our conclusions are applicable for the low-temperature WGS reactions up to 548 K.

In reply to the reviewer, we have added the behaviours of c-Cu and o-Cu in the WGS reaction up to 673 K as Figure S10 on Page S22 in the revised supporting information and discussed the results as the following in the revised manuscript: **“We also evaluated the catalytic activity of c-Cu and o-Cu in the WGS reaction up to 673 K (Figure S10). o-Cu becomes active at 623 K, and c-Cu exhibits the CO conversion that increases with the reaction temperature up to 623 K but then decreases. The recycling measurements of catalytic activity and the SEM images of Cu NCs after the activity evaluation clearly demonstrate that both Cu NCs are not stable and undergo obvious morphological changes during the activity evaluation up to 673 K.”** (Please see Lines 8-14 on Page 8) and **“The above comprehensive characterization results demonstrate the accumulation of formate and hydroxyl intermediates during the WGS reaction up to 548 K on the o-Cu surface but not on the c-Cu and d-Cu surfaces that should lead to the self-poisoning and subsequent inactivity of o-Cu NCs. It is noteworthy that the surface reaction between co-adsorbed formate and hydroxyl to produce CO₂ and H₂O at 540 K observed on the used o-Cu catalyst should be thermodynamically quenched during the WGS reaction with H₂O as a reactant. Meanwhile, the accumulation of formate and hydroxyl intermediates on the o-Cu surface can be alleviated at higher reaction temperatures, as the o-Cu NCs are active in catalyzing the WGS reaction at 623 K and above.”** (Please see Paragraph 2 on Page 11).

Comment 6) On p. 9 the authors state that the CO oxidation takes place on the Cu – Cu suboxide interface. However the data only shows this oxygen species as being active and there is no

evidence for the reaction taking place exclusively at the interface, as opposed to on top of the suboxide. It should be made clearer to the reader that the data presented does not identify if the reaction takes place at the interface or on top of the suboxide regions.

Author Reply: We appreciate the reviewer's insightful comments very much and we have seriously considered them. In the DRIFTS of CO adsorption on Cu NCs at 123 K (Figure 1F), only vibrational features of CO adsorbed at Cu surfaces were observed. This suggests that CO species in CO oxidation at the Cu-Cu_xO interfaces should be CO adsorbed at the Cu surfaces.

In reply to the reviewer, we have clarified this issue as the following in the revised manuscript: "Since CO adsorbs on the metallic Cu surface instead of the Cu_xO surface (Figure 1F), the reaction of Cu_xO with CO on the c-Cu surface should proceed at the Cu-Cu_xO interface." (Please see Lines 8-11 on Page 12).

Comment 7) The interface as the locus of activity for the inverse structures of ZnO/Cu single crystal surfaces has been demonstrated by the work of Rodriguez et al. (not included in the refs. here; ref. 27 reaches the same conclusion for ceria). At the interface, the oxidation state of the active Cu species has been debated in the literature, and convergence to Cu(I) is shown. In the work by Flytzani-Stephanopoulos et al., the WGS reaction has been found to occur on cationic metal species, including Au(I), Pt(II), Cu(I), etc. In other words, the reduction to the metal is not complete. See for example, Rui Si et al., "Structure sensitivity of the low-temperature water-gas shift reaction on Cu-CeO₂ catalysts", *Catal. Today*, **2012**, 180, 68. How to probe the interface with any spectroscopy in the preponderance of metallic Cu as in the authors' NCs is a formidable and perhaps untenable task.

Author Reply: We appreciate the reviewer's insightful comments and visions on the metal-oxide interface active site in the WGS reaction very much. This is a topic of broad interest and great importance, and a lot of nice papers have been published, as demonstrated by the reviewer's comments. We read all these papers when we carried out the project. After obtaining the comprehensive experimental results, we are very excited because our Cu NCs-based catalysts do provide novel insights in the Cu-based WGS catalysts by revealing the important role of Cu facets in the low-temperature WGS reaction. Our results clearly demonstrate that c-Cu NCs enclosed with the {100} facets are very active in catalyzing the WGS reaction up to 548 K while o-Cu NCs enclosed with the {111} facets are inactive. The Cu-Cu suboxide (Cu_xO, $x \geq 10$) interface of Cu(100) surface is the active site on which all elementary surface reactions within the catalytic cycle proceed smoothly. However, the formate intermediate was found stable at the Cu-Cu_xO interface of Cu(111) surface to accumulate and poison the surface at low temperatures. Our results also demonstrate the Copper facet effect is also applicable for the ZnO/Cu catalysts with ZnO/c-Cu more active than ZnO/o-Cu. However, presently we are not sure the copper species at the Cu-ZnO interface. We are going to work on it although it is very challenging and time-consuming, as the reviewer commented. But our advantage is that we can compare the results of ZnO/c-Cu and ZnO/o-Cu with the same composition but different Cu structures to gain the insights. We hope that we will succeed and present another story in the future.

Comment 8) A cautionary statement in the kinetics work. Have the authors conducted their tests

at steady-state and from high-T to low-T? if not, structural changes at the high temperatures, could have caused lower rates, i.e. an erroneous apparent activation energy, which can explain the values of ~ 30 kJ/mol reported for the Cu/ZnO catalysts. There is no reason for the activation energy to be different, if the same cuprous species is the active site in all such catalysts, i.e. the mechanism does not change. Cu should not be compared to other metals like gold. The latter has its own “signature” with E_a lower than that of Cu (or Pt) on any support.

Author Reply: We appreciate the reviewer’s insightful comments very much and we have seriously considered them. As we described, the reported reaction data were all acquired at the steady state. At each temperature, three data were measured at different reaction times and reproducible. As shown in Figure S7, each catalyst was measured for two cycles and the reaction data were also reproducible. We have also done a Weisz–Prater analysis and a Mears analysis proposed by Oyama to confirm the absence of mass and heat transfer limitations under our reaction conditions. And the Arrhenius plots were made using reaction data with CO conversions below 20%. Thus we believe that the reported kinetics in our manuscript is reliable. However, we have accepted the reviewer’s suggestion to evaluate the catalytic performance of c-Cu and ZnO/c-Cu from low temperature to high temperature and then from high temperature to low temperature. The results (shown below) also confirm the reliable kinetic measurements. Thus the lower apparent activation energy of ZnO/c-Cu catalysts than of Cu/Zn/Al₂O₃ and ZnO/o-Cu should be due to the Cu facet effect, although the underlying mechanism needs further studies and is not the topic of the present manuscript. It is likely that the Cu(I) species on Cu(111) could exhibit different reactivity from that on Cu(100); it is also likely that the ZnO on Cu(111) could be different from that on Cu(100). We previously reported that the structure and catalytic activity of the CuO thin film grown on the Cu₂O substrate are strongly affected by the Cu₂O substrate structure (*Angew. Chem. Int. Ed.* **50** (2011): 12294-12298).

In reply to the reviewer, we have added the results of activity evaluation of c-Cu and ZnO/c-Cu from low temperature to high temperature and then from high temperature to low temperature respectively as Figure S7B on Page S19 and Figure S26 on Page S43 in the revised supporting information. We have also described the Weisz–Prater analysis and a Mears analysis in the revised supporting information (Please see Page S6-S8). We have also rewritten the relevant texts as the following in the revised manuscript: **“The c-Cu and d-Cu NCs are also catalytically stable at 548 K (Figure 2B and Figure S7), and the SEM images (Figure S9) demonstrate that the Cu NCs preserve their morphologies after the activity evaluation up to 548 K. Results of a Weisz–Prater analysis and a Mears analysis²⁶ in Supplemental Information show the absence of mass and heat transfer limitations under our reaction conditions.”** (Please see Lines 13-17 on Page 7) and **“Meanwhile, the ZnO/c-Cu catalyst is stable (Figure S26).”** (Please see the last sentence on Page 15). We have also removed the comparison between our catalysts with other metals. We have also re-ordered the references and figures accordingly.

Comment 9) The oxidation of flat copper surfaces has been extensively studied. Although the exact structure of Cu under reaction conditions is not known, the design of the DFT model for the Cu-Cu₂O interface should resemble some of the motifs found on carefully studied single crystal surfaces (Besenbacher and Norskov, *Prog. Surf. Sci.* **1993**, *44*, 5). For example, O on Cu(111) readily reconstructs into ring structures over a wide range of O coverages and temperatures (Matsumoto et al. *Surf. Sci.*, **2001**, *471*, 225; Soon et al. *Phys. Rev. B*, **2006**, *73*, 165424; Yang et al. *J. Phys. Chem. C*, **2010**, *114*, 17042), but a structure like the zig-zag pattern that the authors chose as their model has never been observed experimentally. This is unfortunate given the large library of data on partially oxidized Cu surfaces, and dramatically diminishes the impact of the calculations with the structure chosen. The authors should provide some justification to the reader for the design of their DFT model, and should also provide the reader with information of observed O/Cu structures from the literature, including some of the references listed above, and how their structures differ.

Minor Concerns:

Author Reply: We appreciate the reviewer's insightful comments very much and we have seriously considered them. Actually during our DFT calculations, we built the oxide structure based on ring-like structures, which resemble very much the structures proposed in literatures. However, to simulate the reaction at the interface between Cu and Cu₂O domain but remain computational affordable for exploration of full reaction network, we use a zig-zag chain representing the edge of Cu₂O ring-like structures (Shown below). For simplicity, the lattice constant of Cu₂O ring-like structure was assumed same with Cu metal underneath.

In reply to the reviewer, we have clarified this issue by showing the whole structures in Figure 5A on Page 33 and rewriting the relevant texts with additional references suggested by the reviewer as the following in the revised manuscript: **“The mechanisms of morphology-dependent catalytic performance of Cu NCs in the WGS reaction were further explored by DFT calculations employing Cu(100) and Cu(111) surfaces with Cu₂O islands to respectively model the c-Cu and o-Cu surfaces with Cu_xO (Figure 5A). According to previous results,³⁰⁻³³ ring-like structures were adopted for Cu₂O islands on both Cu(100) and Cu(111) surfaces. This gives a zig-zag chain structure of Cu₂O at the Cu-Cu₂O interface that was used during simulations of reactions at the Cu-Cu₂O interface.”** (Please see the Last paragraph on Page 12). We have also re-ordered the references accordingly. We have also described the calculation methods in very detail in the revised supporting information (Please see Page S8-S10).

Comment 10) Recently there has been a study on Cu single crystals from the Salmeron group showing that the (111) facet has the best activity towards CO oxidation. (Eren et al. *J. Phys. Chem. C* **2015**, *119*, 14669) CO oxidation is a crucial step in the water-gas-shift reaction; could the authors comment on the inverse in the activity hierarchy for these two reactions?

Author Reply: We appreciate the reviewer's insightful comments very much. We have read the paper (denoted as JPC paper thereafter) mentioned by the reviewer very carefully. In the JPC paper the reaction of CO with chemisorbed oxygen on three low-index faces of copper was studied and it was found that on Cu(111) the rate was one order of magnitude faster than that on Cu(100) and two orders of magnitude faster than that on Cu(110). The story is very different from ours. Firstly, only CO+O(a) surface reaction was studied in the paper while we studied the catalytic WGS reaction with a complex surface reaction network. Such a surface reaction rate can not simply represent the catalytic activity. Actually it was reported that among Cu, Cu₂O and CuO, Cu exhibits excellent initial activity in catalyzing CO oxidation, but then facilely deactivates (ChemCatChem, 2011, 3, 24); Secondly, chemisorbed oxygen species on Cu surfaces were studied in the paper while the suboxide species was the active species in our case. Our DFT calculation results (Table S3) show that the single surface reaction of CO_{Cu} + O_{Cu₂O} → CO_{2, Cu₂O} + O_v proceeds with a lower barrier at the Cu(111)-Cu₂O interface (0.33 eV) than at the Cu(100)-Cu₂O interface

(0.56 eV), similar to the CO+O(a) surface reaction reported in the JPC paper. However, in the WGS reaction at the Cu-Cu₂O interface, the H₂ production is the rate limiting step.

Comment 11) I suggest a modification to the title from “the active Cu structure...” to “the most active Cu facet...” because all Cu structures investigated are indeed active, this work is aimed towards finding the most active facets (as opposed to general structure such as nanoparticle size or shape).

Author Reply: We appreciate the reviewer’s nice suggestion and take the pleasure to accept it. The title of our revised manuscript is “**The Most Active Cu Facet for Low-Temperature Water Gas Shift Reaction**”.

Comment 12) The authors may want to consider re-labeling of the “suboxide” species, because it suggests that the oxide is subsurface. However the authors find that there is O on the surface at these suboxide regions, so perhaps calling these sites simply “oxide” would be more fitting.

Author Reply: We appreciate the reviewer’s kind suggestion very much and we have seriously considered them. The species labelled as copper suboxide in our manuscript exhibits a Cu(I) feature and an O feature with an O 1s binding energy at 530.3 eV. The O 1s feature of the suboxide species is very different from that in copper oxides (Cu₂O and CuO). Such a species was previously observed and labelled as copper suboxide in Ref. 14 cited in the manuscript and the paper of Schedel-Niedrig et al. Partial methanol oxidation over copper: active sites observed by means of in situ X-ray absorption spectroscopy. *Phys. Chem. Chem. Phys.* **2**, 3473-3481 (2000), and we then followed the references labelling. The suboxide does not mean subsurface oxide, but means its Cu:O stoichiometric is much larger than copper oxides. Thus we consider that it is appropriate to use copper suboxide instead of copper oxides.

In reply to the reviewer’s comments, we have clarified this issue by defining the suboxide and cited *Phys. Chem. Chem. Phys.* **2**, 3473-3481 (2000) as Ref. 13 in the revised manuscript as the following: “**Cu suboxide (Cu_xO, x≥10)**” (Please see Line 8 on Page 2 and Line 20 on Page 5). We have also re-ordered the references accordingly.

Authors' Reply to Reviewer 3's Comments and Revisions

Comment: This manuscript describes a promising approach to determine the active site structure of Cu-based catalysts for the low temperature water-gas shift (WGS) reaction. The information is then used to prepare an improved catalyst sample. The WGS reaction is of great industrial importance and the currently used catalyst, Cu/ZnO/Al₂O₃ is also Cu-based; thus, this topic is of high relevance and could be appealing to a large audience.

Despite the potential impact of the work, there are several gaps in the argumentation. These gaps are described below and require major revisions before this manuscript can be reviewed again.

Author Reply: We appreciate the reviewer's positive and valuable comments very much. We have seriously considered the reviewer's comments and revised the manuscript accordingly. We hope that the revised manuscript will be suitable for the publication in *Nature Communications*.

Comment: The interpretation of the TPRS data on page 7 leads the authors to claim that H₂ desorption is the rate-limiting step of Cu-catalyzed WGS. I don't see sufficient evidence for this conclusion. If the active site is the Cu/Cu-suboxide interface, CO₂ desorption could simply be a result of Cu-suboxide reduction by CO. I assume similar CO₂ desorption peaks would be observed in the absence of H₂O. Also, if H₂ desorption was rate-limiting, why did the authors not simulate this step with DFT? The adsorption/desorption of H₂ in Table S5 is assumed to be quasi-equilibrated and no barrier is reported.

Author Reply: We appreciate the reviewer's insightful comments very much and we have seriously considered them. In our manuscript, we discussed that, on the basis of TPR results, the H₂ production, not the H₂ desorption commented by the reviewer, is the rate-limiting step of Cu-catalyzed WGS reaction. The barrier of H_{Cu}+H_{Cu}→H₂ was calculated to be 0.90 and 0.85 eV at the Cu₂O-Cu interface respectively on Cu(111) and Cu(100). It can be found from the data listed in Tables S3-S5, the elementary step with the largest barrier among all three reaction mechanisms is O_{Cu2O}H → O_{Cu2O} + H_{Cu} involved in the H₂ production, 0.98 and 1.09 eV at the Cu₂O-Cu interface respectively on Cu(111) and Cu(100). Thus the experimental and DFT calculations results agree with each other.

As demonstrated in the XPS results, Cu suboxide and hydroxyl groups exist at the suboxide-Cu interface on the Cu surface subjected to the WGS reaction, thus the CO₂ production can be due to both the reduction of suboxide by CO and the CO oxidation by hydroxyl groups, as demonstrated by the calculated three reaction mechanisms. We carried the experiments of the CO reduction of Cu NCs activated by water decomposition at 250 °C, and the acquired CO₂ production is similar to those in the CO+H₂O TPRS on Cu NCs but with much reduced intensities (shown below). However, we do not consider that the CO reduction results of Cu NCs activated by water decomposition at 250 °C are important and thus decide not to include them in the manuscript.

In reply to the reviewer’s comments, we have included the calculated barriers of $H_{Cu}+H_{Cu} \rightarrow H_2$ at the Cu_2O-Cu interface respectively on $Cu(111)$ and $Cu(100)$ in Tables S3-S5 in the revised supporting information and clarified the issue as the following in the revised manuscript: **“Figure 3A presents TPRS spectra of WGS reaction over the Cu NCs. The CO₂ production was observed to occur prior to the H₂ production, suggesting the H₂ production as the rate-limiting step in the Cu-catalyzed WGS reaction.”** (Please see Lines 4-6 on Page 9) and **“In all three pathways the elementary surface reaction with the largest activation energy is the reaction of $O_{Cu_2O}H \rightarrow O_{Cu_2O} + H_{Cu}$ involved for the H₂ production. This agrees with the experimental observations that the H₂ production is the rate-limiting step in the Cu-catalyzed WGS reaction (Figure 3A).”** (Please see the texts between Line 21 on Page 13 and Line 2 on Page 14).

Comment: Ultimately, the authors attribute WGS reactivity to the presence of Cu/Cu-suboxide interfaces. With this active site assignment, many of the performance data are questionable. The BET surface area would not be a good way to normalize the TOF. Instead, the authors should normalize by the number of interface sites. The observed differences may simply be a difference in how well the different Cu facets oxidize to a suboxide, or how well the epitaxial relationship between the Cu metal and the suboxide matches. It is likely that the thermodynamically most stable (111) facet is not easily oxidized. From the DFT results, the authors could easily extract the interfacial energy between $Cu(111)$ or $Cu(100)$ and Cu_2O , and determine if there is a significant stability difference.

Author Reply: We appreciate the reviewer’s insightful comments very much and we have seriously considered them. We definitely agree with the reviewer that the catalytic activity should be expressed by the reaction rate normalized to the number of active Cu suboxide-Cu interface site. However, it is a very challenging task and even could be an impossible task to identify the number of active interface site on powder samples. Thus it is usually approximated by the reaction rate normalized to the catalyst mass and more accurately the reaction rate normalized to the number of surface Cu sites. In our case we used the reaction rate normalized to the number of surface Cu sites. The difference in the activity can be demonstrated more evidently using apparent activation energy than using the normalized reaction rate. Our results of apparent activation energy (Figure 2C) clearly show that c-Cu NCs is more active than d-Cu NCs. Experimentally, we observed that c-Cu NCs are active in catalyzing the low-temperature WGS reaction while o-Cu NCs are not. The Cu species on C-Cu and o-Cu surfaces are similar, $Cu(0)$ and

Cu suboxide, although the ratios are different. The comprehensive characterization results demonstrate the accumulation of formate and hydroxyl intermediates during the WGS reaction up to 548 K on the o-Cu surface but not on the c-Cu and d-Cu surfaces that should lead to the self-poisoning and subsequent inactivity of o-Cu NCs. Thus when carrying out the DFT calculations, we focused the identification of the stability of various surface intermediates at the Cu suboxide-Cu interface on Cu(111) and Cu(100) surfaces. To simulate the reaction at the interface between Cu and Cu₂O domain but remain computational affordable to explore full reaction network, we use a zig-zag chain structure representing the edge of Cu₂O ring-like structures. Moreover, the lattice constant of Cu₂O ring-like structure was assumed same with Cu metal underneath for simplicity. Thus we did not consider the Cu suboxide-Cu epitaxial relationships and Cu-Cu₂O interfacial energies commented by the reviewer. During the revision, we estimated the interfacial energy between Cu and Cu₂O and compared the difference between Cu(111) and Cu(100) by constructing and optimizing one layer Cu₂O(111)-(1×1) and Cu₂O(100)-(1×1) epitaxial growth respectively on Cu(111)-(1×1) and Cu(100)(1×1). From the optimized structures, the corresponding Cu-Cu₂O interfacial energies were extracted to be 120 meV/Å² for Cu(100) and 70 meV/Å² for Cu(111), respectively (shown below). Thermodynamically more stable Cu(111) interacts more weakly with oxide film, in line with the larger resistance toward oxidation as pointed by the reviewer. However, since these results are not related with our experimental results much, we considered not to include them in the manuscript.

Comment: Side note: a brief clarification why the BET surface area of larger o-Cu NCs (937 nm) is larger than that of smaller c-Cu NCs (877 nm) would be appropriate. It is likely due to the higher site density of the {111} planes.

Author Reply: We appreciate the reviewer's insightful comments very much and we have seriously considered them. We considered two likely factors giving larger specific BET surface areas of the o-Cu₂O and o-Cu NCs with a larger average edge length respectively than the c-Cu₂O and c-Cu NCs. One is the morphology effect: the surface area-to-volume ratios of an octahedron and a cube can be geometrically calculated respectively as $3\sqrt{6}/a$ and $6/a$ (a being the edge length); the other is the facet effect: the surface atom density is higher on the {111} facets exposed on the octahedral NCs than on the {100} facets exposed on the cubic NCs for both Cu₂O and Cu with cubic phases.

In reply to the reviewer, we have clarified this issue as the following in the revised manuscript: **“Although with a larger average edge length, the o-Cu₂O and o-Cu NCs exhibit larger specific BET surface areas respectively than the c-Cu₂O and c-Cu NCs due to the morphology and facet effects. The surface area-to-volume ratios of an octahedron and a cube can be geometrically calculated respectively as $3\sqrt{6}/a$ and $6/a$ (a being the edge length); meanwhile, the surface atom density is higher on the {111} facets exposed on the octahedral NCs than on the {100} facets exposed on the cubic NCs for both Cu₂O and Cu with cubic phases.”** (Please see Lines 4-10 on Page 5).

Comment: The high surface roughness is a concern. A rough surface will necessarily have defect and step sites. Were there any efforts to quantify the amount of surface defects? The CO vibration experiments should exhibit signals of CO molecules strongly adsorbed on steps/defects. Did the authors try to compare the signal intensities to get a rough estimate of the ratio of step to terrace sites?

Author Reply: We appreciate the reviewer’s insightful comments very much and we have seriously considered them. The vibrational frequency of CO adsorbed on Cu surfaces is sensitive to the coordination of Cu sites. It was reported that CO adsorbed at stepped sites and defects on Cu surfaces exhibits vibrational features above 2100 cm⁻¹ (Dulaurent, O., Courtois, X., Perrichon, V., Bianchi, D. Heats of adsorption of CO on a Cu/Al₂O₃ catalyst using FTIR spectroscopy at high temperatures and under adsorption equilibrium conditions. *J. Phys. Chem. B* **104**, 6001-6011 (2000)). In Figure 1F, the absence of vibrational features above 2100 cm⁻¹ suggests the low density of stepped sites and defects on the surfaces of all acquired Cu NCs.

In reply to the reviewer, we have clarified this issue by discussing as the following with citing *J. Phys. Chem. B* **104**, 6001-6011 (2000) as Ref. 25 in the revised manuscript: **“The absence of vibrational features above 2100 cm⁻¹ that correspond to CO adsorbed at stepped sites and defects on Cu surfaces²⁶ also suggests the low density of stepped sites and defects on the surfaces of all acquired Cu NCs.”** (Please see Lines 18-21 on page 6). We have also re-ordered the references accordingly.

Comment: The DFT results suggest that there is no significant difference between Cu(111) and Cu(100) model interfaces with Cu₂O present. The only difference was speculated to be the formate coverage based on the formation and decomposition barriers. Such speculation is rather unreliable and often fails, even in simple cases. The coverage of formate depends on the coverage of H and CO₂, which in turn depend on everything else. Also, if formate is a spectator species, it will be formed until it reached its saturation coverage. At this point the formation and decomposition barriers are no longer relevant to estimate the formate coverage at the interface under reaction conditions. Hence, I see value in the DFT work to assign the active site to the Cu/Cu₂O interface, but I do not see agreement with the experimentally observed activity increase on the (100) facet.

Author Reply: We appreciate the reviewer’s insightful comments very much and we have seriously considered them. Comprehensive experimental results demonstrate (1) the obvious facet effect of Cu particles in the low-temperature WGS reaction with the most active Cu{100}

facet and the inactive Cu{111} facet; (2) the H₂ production as the rate-limiting step in the Cu-catalyzed WGS reaction; (3) the accumulation of formate and hydroxyl intermediates during the WGS reaction up to 548 K on the o-Cu surface but not on the c-Cu and d-Cu surfaces that should lead to the self-poisoning and subsequent inactivity of o-Cu NCs; (4) the Cu-Cu_xO interface on the Cu surface capable of dissociating water and reacting with CO as the active site to catalyze the WGS reaction. Employing Cu(111) and Cu(100) surfaces with Cu₂O as the model, the DFT calculation results demonstrate (1) the much more facile water activation at the Cu₂O-Cu interface than at the Cu surface and the more facile water dissociation at the Cu-Cu₂O interface into OH_{Cu} and O_{Cu₂O}H on Cu(100) than on Cu(111); (2) proceeding of the WGS reaction via three pathways at the Cu-Cu₂O interface of both Cu(100) and Cu(111) surfaces with similar largest activation energies of around 1 eV; (3) the reaction of O_{Cu₂O}H → O_{Cu₂O} + H_{Cu} involved for the H₂ production with the largest activation energy in all three reaction pathways; (4) the large barrier for the HCOO_{Cu₂O} decomposition during the low-temperature WGS reaction at the Cu-Cu₂O interface of Cu(111) surface (1.47 eV) but not at the Cu-Cu₂O interface of Cu(100) surface (0.95 eV) that should lead to the accumulation of the formate species at the Cu-Cu₂O interface of Cu(111) surface. It can thus be seen that the DFT calculation results qualitatively agree with our experimental results quite well. It is noteworthy that all our results and discussions are restricted within the low-temperature WGS reaction up to 547 K, as we repeatedly emphasize in the manuscript. Thus the formate species with decompositions barriers similar to the barriers of elementary surface reactions with the WGS reaction can reach an equilibrium with CO₂ and H, and will not accumulate at the Cu₂O-Cu interface; however, the HCOO_{Cu₂O} species at the Cu-Cu₂O interface of Cu(111) surface exhibits a decomposition barrier of 1.47 eV much larger than the barriers of elementary surface reactions with the WGS reaction, thus it can not reach an equilibrium with CO₂ and H, and will accumulate at the Cu₂O-Cu interface of Cu(111) surface with the WGS reaction proceeding and eventually poison the surface.

In reply to the reviewer, we have made the revisions in the revised manuscript as the following:

- 1) We have moved the Cu LMM AES and O 1s XPS results of water activation on Cu NCs from the supporting information to the revised manuscript as Figure 4 on Page 32 in order to demonstrate the facile water activation on c-Cu than on d-Cu and o-Cu and to provide experimental results in accordance with the DFT calculation results shown in Figure 5. We have re-ordered all figures in the revised manuscript.
- 2) We have described how the DFT calculation results agree with the experimental results in the contexts of the revised manuscript as the following: **“Thus the water activation is much more facile at the Cu₂O-Cu interface than at the Cu surface, and the water dissociation at the Cu-Cu₂O interface into OH_{Cu} and O_{Cu₂O}H should proceed more facilely on Cu(100) than on Cu(111). These agree with the above experimental observations (Figure 4).”** (Please see Lines 11-14 on page 13); **“All three pathways can proceed at the Cu-Cu₂O interface of both Cu(100) and Cu(111) surfaces with similar largest activation energies of around 1 eV. In all three pathways the elementary surface reaction with the largest activation energy is the reaction of O_{Cu₂O}H → O_{Cu₂O} + H_{Cu} involved for the H₂ production. This agrees with the experimental observations that the H₂ production is the rate-limiting step in the Cu-catalyzed WGS reaction (Figure 3A).”** (Please see the texts between Line 20 on Page 13 and Line 2 on Page 14); **“Thus, it can be expected that the formate species will accumulate at the Cu-Cu₂O interface of Cu(111) surface due to the large barrier for the HCOO_{Cu₂O}**

decomposition during the low-temperature WGS reaction but not at the Cu-Cu₂O interface of Cu(100) surface, which will eventually block the active Cu-Cu₂O interface of Cu(111) surface. This is consistent with the experimental observations that the c-Cu catalyst is active in catalyzing the low-temperature WGS reaction up to 548 K while the o-Cu catalyst is not and is with accumulated formate species on the surface (Figures 2 and 3).” (Please see Lines 11-18 on page 14).

Comment: Were the transition states found in the NEB method verified by vibrational analysis showing a single imaginary mode? I would be beneficial if the authors clearly stated that zero point energies and entropy corrections were neglected.

Author Reply: We appreciate the reviewer’s kind suggestions very much and accept it. We have clarified the issue as the following in the revised supporting information: “**The transition states were verified by vibrational analysis showing a single imaginary mode. Zero point energies and entropy corrections were neglected in DFT calculations.**” (Please see Lines 10-12 on Page S10).

Comment: Could the authors please clarify if the reaction energies and barriers are taken with respect to isolated reactants/products? The equation in the SI suggests this but an explicit statement is preferred. If energies are reported at infinite separation, could the authors also include the interaction energies for the co-adsorbed states that form the initial and final states of their NEB calculations?

Author Reply: We appreciate the reviewer’s kind suggestions very much and have accept it. We have clarified the issue as the following in the revised supporting information: “**The activation energies of elementary reactions (E_a) and reaction energies of elementary reactions (E_r) were taken with respect to isolated reactants/products.**” (Please see Lines 12-14 on Page S10).

In order to address the reviewer’s comments on the interaction energy, four typical elementary surface reactions were selected to calculate the interaction energy between isolated species and co-adsorbed states. The results are given in the Table below. It can be found that most of them have modest interaction energies (less than 0.09 eV), and therefore can be neglected. The small interaction might come from the relative large supercell (2×4) used in present work. Large interaction energies were observed for water dissociation at Cu₂O-Cu interfaces, being about 0.47 eV for Cu₂O-Cu(111) and 0.41 eV for Cu₂O-Cu(100). This comes from the cost of hydrogen bond between dissociated fragments at the co-adsorbed states.

Table. The Interaction Energies (eV) for the Co-adsorbed States in Selected Elementary Reactions.

Elementary Surface Reactions	States	Interaction Energy	
		(111)	(100)
$\text{H}_2\text{O}_{\text{Cu}} + \text{O}_{\text{Cu}_2\text{O}} \rightarrow \text{OH}_{\text{Cu}} + \text{O}_{\text{Cu}_2\text{O}}\text{H}$	Final State $\text{OH}_{\text{Cu}} + \text{O}_{\text{Cu}_2\text{O}}\text{H}$	0.47	0.41
$\text{HCOO}_{\text{Cu}_2\text{O}} \rightarrow \text{CO}_{2, \text{Cu}_2\text{O}} + \text{H}_{\text{Cu}}$	Final State $\text{CO}_{2, \text{Cu}_2\text{O}} + \text{H}_{\text{Cu}}$	0.05	0.08
$\text{H}_{\text{Cu}} + \text{H}_{\text{Cu}} \rightarrow \text{H}_2(\text{g})$	Initial State $\text{H}_{\text{Cu}} + \text{H}_{\text{Cu}}$	0	0

$\text{CO}_{\text{Cu}} + \text{O}_{\text{Cu}_2\text{O}}\text{H} \rightarrow \text{COOH}_{\text{Cu}_2\text{O}}$	Initial State $\text{CO}_{\text{Cu}} + \text{O}_{\text{Cu}_2\text{O}}\text{H}$	0.07	-0.02
--	---	------	-------

In reply to the reviewer, we have clarified this issue in the revised supporting information (Please see the texts and Table between Line 14 on Page S10 and Line 13 on Page S11).

Comment: The manuscript must be edited for grammar and spelling. Most importantly “morphology-reserved” should probably be “morphology-preserved” .

Author Reply: We appreciate the reviewer’s careful reading and kind suggestions very much and have replaced “reserved” with “preserved” throughout the revised manuscript and supporting information. We have also done our best to polish the English of the revised manuscript and supporting information.

Reviewers' comments:**Reviewer #1 (Remarks to the Author):**

The authors have thoroughly address most of the extensive questions from the reviewers. The manuscript has improved considerably and I recommend now its publication.

Reviewer #2 (Remarks to the Author):

The authors have provided satisfactory answers to most of my questions. However, they have not addressed comment 3 convincingly. This was:

3)On p. 7 the authors discuss that both the Cu and Cu₂O NCs are equivalent in activity under reaction conditions. The authors then state this a evidence for Cu(0) as the active site, as opposed to Cu(I). However, this statement is inaccurate and should be removed, as the authors go on to show how a partially oxidized surface is much more active than the bare metal. I would add to this that at the interface with ZnO, Cu(I) is the active site, as supported by the voluminous literature on the subject. Thus, a unified mechanism for the WGS reaction on copper does not result from all the cases examined here. This can be achieved in my view, and should be the focus of a second revision/analysis that I strongly recommend the authors to undertake. During reaction, Cu(I) species may be created on all Cu surfaces. The Cu₂O/Cu interface and the ZnO/Cu(I) interfacial species are left to be different in the discussion. The apparent activation energies are also very different. Why is that so? Have the authors considered the possibility of a mechanism switch? different on Cu metal surfaces (and with lower activity) than on Cu(I)sites at the interface with ZnO (or another oxide)? that can explain the disparate activities of the two catalysts.

Reviewer #3 (Remarks to the Author):

I have read through the 24 page rebuttal letter and I believe that the authors have provided sufficient information to address all comments raised by the reviewers. After the major revision of the manuscript and the inclusion of additional data and references in the text and supplementary information I consider this manuscript publishable in Nature Communications.

Authors' Reply to Reviewer 1's Comments and Revisions

Comment: The authors have thoroughly address most of the extensive questions from the reviewers. The manuscript has improved considerably and I recommend now its publication.

Author Reply: We appreciate the reviewer's recommendation very much. We believe that our paper will be of strong impact in the fields of heterogeneous catalysis and surface chemistry after the publication in *Nature Communications*.

Authors' Reply to Reviewer 2's Comments and Revisions

Comment: The authors have provided satisfactory answers to most of my questions. However, they have not addressed comment 3 convincingly.

Author Reply: We appreciate the reviewer's positive comments on our revised manuscript very much. We have seriously considered the reviewer's comments and further revised the manuscript accordingly. We hope that the revised manuscript will be suitable for the publication in *Nature Communications*.

Comment: On p. 7 the authors discuss that both the Cu and Cu₂O NCs are equivalent in activity under reaction conditions. The authors then state this as evidence for Cu(0) as the active site, as opposed to Cu(I). However, this statement is inaccurate and should be removed, as the authors go on to show how a partially oxidized surface is much more active than the bare metal.

Author Reply: We appreciate the reviewer's comments very much. We agree with the reviewer that Cu nanocrystals with a partially oxidized surface are active in catalyzing low-temperature WGS reaction and the Cu{100} facet is the most active facet. Such an argument is adequately supported by the observed equivalent catalytic activity of the Cu and Cu₂O NCs in under reaction conditions and the structural characterization results. In the revised manuscript we have rewritten the relevant sentence as the following: **"These results demonstrate that Cu nanocrystals with the coexisting Cu(0) and Cu(I) species on the surface are active in catalyzing low-temperature WGS reaction and further support that the Cu{100} facet is the most active facet."** (please see line 22 on Page 8 – Line 2 on page 9). We believe such a description is appropriate.

Comment: I would add to this that at the interface with ZnO, Cu(I) is the active site, as supported by the voluminous literature on the subject. Thus, a unified mechanism for the WGS reaction on copper does not result from all the cases examined here. This can be achieved in my view, and should be the focus of a second revision/analysis that I strongly recommend the authors to undertake. During reaction, Cu(I) species may be created on all Cu surfaces. The Cu₂O/Cu interface and the ZnO/Cu(I) interfacial species are left to be different in the discussion. The apparent activation energies are also very different. Why is that so? Have the authors considered the possibility of a mechanism switch? different on Cu metal surfaces (and with lower activity) than on Cu(I)sites at the interface with ZnO (or another oxide)? that can explain the disparate activities of the two catalysts.

Author Reply: We appreciate the reviewer's insightful comments very much to encourage us to discuss the different active structure and reaction mechanism between Cu NCs and ZnO/Cu-NCs catalysts. We have seriously considered these comments and take the pleasure to accept it. The calculated apparent activation energies show that the copper-ZnO interface of ZnO/Cu NCs catalysts should be intrinsically more active than the Cu-CuO_x interface of corresponding Cu NCs catalysts. We are now working hard to understand the underlying mechanisms, but due to the complex structure of ZnO/Cu-NCs catalysts (the presence of Cu(0), Cu(I) and the ill-defined ZnO structure), we can not provide any discussion on the WGS reaction mechanism over ZnO/Cu NCs

catalysts. However, a very preliminary observation indicates that surface oxidation extent of Cu in Cu-ZnO-based catalysts in the WGS reaction should depend on the Cu particle size.

In the revised manuscript, we have rewritten the relevant texts as the following: **“It can be seen that both Cu(0) and Cu(I) exist on the surface of Cu NCs in ZnO/Cu catalysts. The ZnO/Cu NCs catalysts are much more active than the corresponding Cu NCs catalysts in catalyzing the WGS reaction (Figures 7E and S25E), and the calculated apparent activation energy of ZnO/c-Cu (32.4 ± 0.8 kJ·mol⁻¹) (Figure 7F) and ZnO/o-Cu (55.9 ± 3.9 kJ·mol⁻¹) (Figure S25F) catalysts are much smaller than the corresponding Cu NCs catalysts. These results suggest that the copper-ZnO interface in ZnO/Cu NCs catalysts exhibits much higher intrinsic activity than the Cu-Cu_xO interface in Cu NCs catalysts. However, the detailed mechanism needs further study. Meanwhile, the ZnO/c-Cu catalyst is stable (Figure S26).”** (please see Line 19 on Page 15 to Line 5 on Page 16) and **“The ZnO/o-Cu catalyst exhibits a similar apparent activation energy to the commercial Cu/ZnO/Al₂O₃ catalyst but much larger than that of ZnO/c-Cu catalyst. Therefore, the active structure of ZnO/c-Cu catalyst should be intrinsically more active than those of ZnO/o-Cu and commercial Cu/ZnO/Al₂O₃ catalysts. This demonstrates that the Cu structure in the Cu-ZnO based catalysts plays a key role in determining the catalytic activity in the WGS reaction and the ZnO/c-Cu catalyst with the Cu{100} structures is a highly efficient catalyst. Meanwhile, it could also be inferred that the copper structure of commercial Cu/ZnO/Al₂O₃ catalyst should be dominated by the Cu{111} structures and its catalytic activity can be improved by engineering the Cu structure from the dominant Cu{111} structure into the Cu{100} structure.”** (please see Lines 13-23 on Page 16). We believe such description and discussion are appropriate.

Authors' Reply to Reviewer 3's Comments and Revisions

Comment: I have read through the 24 page rebuttal letter and I believe that the authors have provided sufficient information to address all comments raised by the reviewers. After the major revision of the manuscript and the inclusion of additional data and references in the text and supplementary information I consider this manuscript publishable in *Nature Communications*.

Author Reply: We appreciate the reviewer's recommendation very much. We believe that our paper will be of strong impact in the fields of heterogeneous catalysis and surface chemistry after the publication in *Nature Communications*.

REVIEWERS' COMMENTS:

Reviewer #2 (Remarks to the Author):

There's not much more the authors can do to provide and support a mechanistic argument. I leave it to the editors' discretion the decision to accept this manuscript.